



# Towards understanding the mechanisms of new particle formation
# in the Eastern Mediterranean

Rima Baalbaki[1], Michael Pikridas[2], Tuija Jokinen[1], Tiia Laurila[1], Lubna Dada[1], Spyros Bezantakos[2], Lauri
Ahonen[1], Kimmo Neitola[1,2], Anne Maisser[2], Elie Bimenyimana[2], Aliki Christodoulou[2,3], Florin Unga[2],
Chrysanthos Savvides[4], Katrianne Lehtipalo[1,5], Juha Kangasluoma[1], George Biskos[2], Tuukka Petäjä[1], Veli-
Matti Kerminen[1], Jean Sciare[2], Markku Kulmala[1]

[1]Institute for Atmospheric and Earth System Research (INAR) / Physics, Faculty of Science, University of
Helsinki, P.O. Box 64, Helsinki, 00014, Finland
[2]Climate & Atmosphere Research Centre (CARE-C), The Cyprus Institute, P.O. Box 27456, Nicosia, CY-
1645, Cyprus
[3]IMT Lille Douai, Université de Lille, SAGE - Département Sciences de L'Atmosphère et Génie de
L'Environnement, 59000, Lille, France
[4]Ministry of Labour, Welfare and Social Insurance, Department of Labour Inspection (DLI), Nicosia, Cyprus
[5]Finnish Meteorological Institute, Helsinki, Finland

*Correspondence to: rima.baalbaki@helsinki.fi*

**Abstract**

To quantify the contribution of new particle formation (NPF) to ultrafine particle number and CCN budgets,
one has to understand the mechanisms that govern NPF in different environments and its temporal extent.
Here, we study NPF in Cyprus, an Eastern Mediterranean country located at the crossroads of three continents.
We performed one-year continuous measurements of aerosol particles down to ~ 1 nm in diameter, for the first
time in the Eastern Mediterranean and Middle East (EMME) region. These measurements were complemented
with trace gas data, meteorological variables and retroplume analysis. We show that NPF is a very frequent
phenomenon at this site and has higher frequencies of occurrence during spring and autumn. NPF events were
both of local and regional origin, and the local events occurred frequently during the month with the lowest
NPF frequency. Some NPF events exhibited multiple onsets, while others exhibited particle apparent shrinkage
in size. Additionally, NPF events were observed during the night-time and during episodes of high desert dust
loadings. Particle formation rates and growth rates were comparable to those in urban environments, although
our site is a rural one. Meteorological variables and trace gases played a role in explaining the intra-monthly
variability of NPF events, but did not explain why summer month had the least NPF frequency. Similarly, pre-
existing aerosol loading did not explain the observed seasonality. The month with the least NPF frequency
were associated with higher $H_2SO_4$ concentrations but lower $NO_x$ concentration, which is an indicator of
anthropogenic influence. Air masses arriving from the Middle East were not observed during these month,
which could suggest that precursor vapors important for nucleation and growth are transported to our site from
the Middle East. Further comprehensive measurements of precursor vapors are required to prove this
hypothesis.

# 1   Introduction

Atmospheric new particle formation (NPF) is the process by which oxidized precursor gases initially form
molecular clusters that then further grow in size by multi-component condensation (Kulmala et al., 2014). A
multitude of research studies have focused on this phenomenon over the past two decades because it is a large
source of the global aerosol particle number and cloud condensation nuclei (CCN) load (Gordon et al.,



2017;Merikanto et al., 2009;Pierce and Adams, 2009;Wang and Penner, 2009;Yu and Luo, 2009;Kerminen et al., 2012;Spracklen et al., 2006;Spracklen et al., 2008). Owing to the complex nature and non-linearity of atmospheric processes, studies on NPF in the literature include atmospheric observations (e.g. Kulmala et al., 2013;Ehn et al., 2014;Bianchi et al., 2016;Yao et al., 2018;Williamson et al., 2019;Baccarini et al., 2020;Dall'Osto et al., 2018), chamber experiments (e.g. Sipilä et al., 2010;Tröstl et al., 2016;Wang et al., 2020;Lehtipalo et al., 2016;Kirkby et al., 2011), and theoretical computational studies (e.g. Kurten et al., 2008;Riipinen et al., 2011;Olenius and Riipinen, 2017). The collective scientific outcome from these studies is essential to understand the mechanisms and characteristics of NPF (Kerminen et al., 2018;Lee et al., 2019;Chu et al., 2019) and how it affects the global climate (e.g. Spracklen et al., 2006;Gordon et al., 2017).

The frequency, strength, and spatio-temporal extent of NPF are mainly governed by three factors: the prevailing meteorological conditions, the availability of gaseous precursors and the pre-existing concentrations of aerosol particles (Kerminen et al., 2018;Lee et al., 2019;Nieminen et al., 2018). These atmospheric conditions differ in space and time. Atmospheric conditions are distinct over the Mediterranean basin, especially over the Eastern Mediterranean and Middle East (EMME). This region has been identified as a hotspot for atmospheric and climate change research (Lelieveld et al., 2016;Giorgi and Lionello, 2008). It is surrounded by three continents and is affected by continental, maritime and desert-dust pollution sources (Lelieveld et al., 2002). The surrounding complex orography of the Mediterranean affects atmospheric dynamics and boundary layer processes on different scales (Kostopoulou and Jones, 2007b, a). Further, the dry and hot weather throughout most of the year, with strongly increasing heat extremes, enables intense photochemistry (Lelieveld et al., 2016).

NPF studies over the Mediterranean have focused on the north western basin (Petäjä et al., 2007;Cusack et al., 2013;Berland et al., 2017;Carnerero et al., 2018;Rose et al., 2015;Brines et al., 2015;Hamed et al., 2007;Laaksonen et al., 2005;Casquero-Vera et al., 2020), whereas NPF studies in the eastern basin have been conducted mainly in Greece (Petäjä et al., 2007;Berland et al., 2017;Kalivitis et al., 2015;Kalivitis et al., 2019;Pikridas et al., 2012;Kalkavouras et al., 2019;Kalkavouras et al., 2020;Kopanakis et al., 2013;Siakavaras et al., 2016;Kalkavouras et al., 2017) and very recently in Cyprus (Brilke et al., 2020;Debevec et al., 2018) and Jordan (Hussein et al., 2020). These studies include both short-term campaigns and long-term observation.

Based on long-term measurements, the annual frequency of NPF over the Mediterranean varies between 10 and 36% (Hussein et al., 2020;Kalivitis et al., 2019;Kalkavouras et al., 2020;Kopanakis et al., 2013). The seasonal cycle has a typical maximum during spring (Kalkavouras et al., 2020;Kopanakis et al., 2013;Kalivitis et al., 2019;Pikridas et al., 2012), even though in some urban background sites the highest frequency was observed during summer (Hussein et al., 2020;Hamed et al., 2007). NPF was associated with a high increase in nucleation mode particles in most of the studies. For instance, Carnerero et al. (2018) showed that the impact of NPF on ultrafine particles is much higher than that of traffic near the highly polluted city center of Madrid. The condensation sink, which is a measure of the pre-existing aerosol surface area, was reported to be lower during NPF events in Po valley, Corsica and Crete (Hamed et al., 2007;Berland et al., 2017;Pikridas et al., 2012), while NPF proceeded under both clean and polluted conditions in Barcelona (Cusack et al., 2013), Marseille and Athens (Petäjä et al., 2007). The effect of meteorological conditions on NPF occurrence varied among studies. Simultaneous NPF events were observed in several stations, illustrating that the spatial extent of NPF events can vary from tens of kilometers (Carnerero et al., 2018) to several hundred kilometers (Kalkavouras et al., 2017;Kalkavouras et al., 2020;Berland et al., 2017;Rose et al., 2015;Casquero-Vera et al., 2020). In the Po valley, the production of CCN from NPF was estimated to be comparable to that originating from primary sources (Laaksonen et al., 2005). Similarly, NPF was associated with a strong increase in CCN concentrations in Finokalia and Santorini (Kalkavouras et al., 2019;Kalkavouras et al., 2017;Kalivitis et al., 2015). However, the impact of the increased CCN concentrations on cloud droplet number was shown to be



limited by water availability (Kalkavouras et al., 2017). In Cyprus, mainly in Paphos, Gong et al. (2019)
observed several NPF events where newly-formed particles grew into the CCN size range, with NPF events
being observed on 9 out of 27 measurement days during April 2017  (Brilke et al., 2020). At a more inland
site, NPF was observed on 14 out of 20 days of measurements on March 2015 (Debevec et al., 2018). Since
these studies were less than a month long, further comprehensive measurements are required to unveil the role
of NPF in the atmospheric processes taking place in the EMME region.

The aim of this study is to characterize the seasonal cycle of new particle formation events in the less
represented area of the EMME region. Our measurements were conducted at a rural background site on the
island of Cyprus which lies at the crossroads of three continents in the Eastern Mediterranean. We report the
first long-term analysis of particle number size distribution in the area, down to sizes where the initial
formation occurs. We further explore the role of sulfuric acid, which is the key gas phase precursor for cluster
formation, and other atmospheric variables in initiating NPF at this site.



## 2    Materials and Methods

### 2.1 Measurement site

Cyprus is an island country in the Eastern Mediterranean. It is the most populated island in the Mediterranean Sea, and the third largest in size with an area of 9251 km$^2$. Cyprus is surrounded by Turkey from the north, Syria and Lebanon from the east, Egypt from the south, and Greece from the west/northwest (Figure 1). The measurements reported in this work were carried out at the Cyprus Atmospheric Observatory (CAO; Sciare, 2016), which is a rural background station that operates under the co-operative programme for monitoring and evaluation of the long-range transmission of air pollutants in Europe (EMEP) and the European Research Infrastructure for the observation of Aerosol, Clouds and Trace Gases (ACTRIS) networks, while at the same time it is a designated regional Global Atmospheric Watch (GAW) station. The CAO site (35.038692° N, 33.057850 ° E) is located close to the villages of Agia Marina (≈630 inhabitants) and Xyliatos (≈150 inhabitants) and has an elevation of 532 m above sea level. The proximity of the site is surrounded by vegetation mainly oak and pine trees and Maquis shrubland as it lies at the northern-eastern foothills of the Troodos Mountains. Agriculture areas surround the site from the north direction and are approximately 4 km away. The nearest main urban agglomeration is at least 35 km away. Therefore, it is distant from any major pollution sources, excluding some limited traffic to reach the nearby Forestry Department premises.

The weather at CAO is characterized by hot dry summers and mild, rainy winters. The daily mean temperature is ~19°C and ranges between 1 and 36°C, the daily mean relative humidity is ~55% and ranges between 13 and 82%, and the daily mean ozone level is ~48 ppb, ranging between 26 and 77 ppb (Kleanthous et al., 2014). The most common (> 65% occurrence) wind pattern reaching the site is the northerly "Etesian" winds transporting pollutants from both Europe and Turkey, but more frequently from mainland of Turkey (Pikridas et al., 2018).The remaining air masses originate from North Africa, the Middle East and westerlies air masses that spend several days above the sea before reaching Cyprus. The variable air mass origins at CAO from three different continents allow a representative description of NPF processes for the EMME region as a whole.

[Figure 1 goes here]

### 2.2 Instrumentation

#### 2.2.1    Aerosol particle number size distribution

The particle number size distribution between 1 and 700 nm was determined by combining data from three instruments: an Airmodus A11 Nano Condensation Nucleus Counter (nCNC) system (Vanhanen et al., 2011), a Neutral cluster and Air Ion Spectrometer (NAIS Model 1; Manninen et al., 2016;Mirme and Mirme, 2013), and a Scanning Mobility Particle Sizer (SMPS Model TSI 3080; Wang and Flagan, 1990). The first two instruments were operated at the site for a period of one year from January 27, 2018 to January 26, 2019 while the SMPS measurement period was from January 27, 2018 to November 1, 2018. The monthly availability of data from each instrument is shown in Table S1.

The A11 nCNC is composed of a Particle Size Magnifier (PSM; Airmodus A10) and a Condensation Particle Counter (CPC; Airmodus A20). The overall length of the inlet sampling tube was 60 cm. The PSM was operated in a scanning saturator flow mode between 0.1 and 1.3 liter per minute (lpm), corresponding to a cut-off diameter range of approximately 1.1 to 2.5 nm. It was equipped with an inlet system that performs background (zero) measurements three times a day at random time intervals, and  a core sampling piece for minimizing line losses of sub-3 nm particles (Figure S1). The duration of the background measurements was set to 12 minutes, which is equivalent to 3 full size scans. From June 2018 onwards, the PSM was additionally equipped with a diluter to reduce the humidity of the sampled air. This procedure was necessary because the water content of the air at the measurement site was too high, and it affected the activation efficiency inside



the CPC and therefore distorted the size distribution measurements for the smallest sizes. Further information
about the diluter design and operation can be found in the supplementary information (SI) Sect. 2.2.

The NAIS is a mobility spectrometer designed to determine the number size distribution of ions in the mobility
diameter range of 0.8 – 42 nm, as well as total (naturally charged and neutral) aerosol particles in the mobility
diameter range of ~2 – 42 nm. The instrument operates at the flow rate of ~54 lpm. The length of the NAIS
sampling tube was 65 cm, with an inner diameter of 30 mm.

The SMPS used in this study was composed of a TSI 3081 long Differential Mobility Analyzer (DMA) and a
TSI 3025a CPC. It was operated to measure the aerosol particle size distribution between 15 and 740 nm. The
aerosol and sheath flow were checked weekly and were set to 0.3 and 3 lpm, respectively.  The SMPS was
sampling using an 80-cm long vertical inlet. Drying was achieved using a short nafion dryer, and charge
neutralization was achieved by a GRIMM 5522-A, Americium-241, bipolar neutralizer.

### 2.2.2   Ancillary measurements

Complementary meteorological data (temperature, relative humidity, solar radiation, rainfall, pressure, wind
speed and wind direction) were measured at an elevation of 10 m from the ground at the nearby village of
Xyliatos (35.0140917 N, 33.0492028 E), 2.85 km from the measurement site. Air pollutants (ozone, carbon
monoxide, nitrogen oxides, sulfur oxide, $PM_{10}$ and $PM_{2.5}$) were measured at the collocated EMEP station ~20
158 m from the main measurement container. Additional details about the set-ups and the instrument used can be
found in Kleanthous et al. (2014) and Pikridas et al. (2018).

### 2.3 Data handling

*nCNC:* The scanning nCNC data was inverted into a size distribution with the Kernel inversion method
presented by Lehtipalo et al. (2014), but using customized kernels which follow the instrument specific
detection efficiency calibration curves. The choice of the inversion method was made after a comprehensive
comparison between the Kernel method and the Expectation and Minimization (EM) method (Cai et al.,
2018;Chan et al., 2020). Additional details about the comparability of the two methods and the utilized
inversion parameters are presented in Sect. 2.3 of the SI. After inversion, the data were further corrected for
line losses using the method suggested by Fu et al. (2019) for the sampling line downstream of the core
sampling inlet, and using the Gormley and Kennedy equation for the line losses inside the 6-cm-long core
sampling piece (Gormley and Kennedy, 1948).

*NAIS:* The NAIS data were inverted with the instrument specific algorithm (done by the NAIS SPECTOPS
software). The data were later corrected for line losses using the Gormley and Kennedy equation for laminar
flow (Gormley and Kennedy, 1948). It is essential to note that the flow through the sampling inlet of the NAIS
actually lies in the transient regime (Re = 2376), however the penetration efficiency using this inlet was
comparable for laminar flow and turbulent flow (calculated using the equation of turbulent inertial deposition
from Brockmann (2011)), thus we used the correction based on laminar flow (Figure S3).

*SMPS:* The data from the SMPS were inverted using the TSI's Aerosol Instrument Manager (AIM, version
9.0) software. Afterwards, line loss correction was applied using Gormley and Kennedy equation. Additional
corrections based on lab calibrations were also applied to account for the low CPC detection efficiency below
15 nm.

*Full Particle Size Distribution (PSD):* The data from the three particle sizing instruments were used to
reconstruct the full particle size distribution between 1.1 and 736 nm (nCNC: 1.1 to 2.4 nm; NAIS particle
mode: 2.4 to 30 nm; SMPS: 30 to 736 nm). However, since the NAIS is known to overestimate concentrations
in particle mode, the overlapping measurement range with the SMPS was used to further correct the NAIS





data assuming that the NAIS overestimate concentrations uniformly over the whole measurement range, which
is a reasonable assumption for old NAIS models based on calibration results (Gagné et al., 2011;Kangasluoma
et al., 2020). Additionally, the SMPS measured dry aerosol particle number distributions, which can differ
considerably from the ambient aerosol particle number size distribution. Thus, we back-calculated the
distribution at ambient conditions from the dry distribution using the hygroscopicity model of Petters and
Kreidenweis (2007) and mean kappa values. Additional information about these calculations and its effect on
sink calculations are presented in Sect. 4 of the SI material. The full PSD using the distribution at ambient
conditions was reconstructed up to 1500 nm. This does not imply that the measurement range was extended to
1500 nm, but rather that now we account for particles that were originally of sizes up to 1500 nm but were
dried to sizes below 740 nm in the SMPS sampling line. Finally, the PSD data was run through a 2D median
filtering algorithm with a 3-by-3 neighborhood window. Moreover, the data was manually checked for the
success of the outlier and noise removal techniques.

*Complementary data:* Gas and meteorology data sets were run through an outlier removal algorithm and
filtered for erroneous samples. The outlier detection method was based on removing data points that are more
than three standard deviations from a moving median (Davies and Gather, 1993;Pearson et al., 2016).

**2.4 Event classification**
The reconstructed full particle size distribution daily plots were used to categorize measurement days into NPF
event days, non-event days and undefined days based on a classification that combines the schemes reported
in literature (Dal Maso et al., 2005;Hirsikko et al., 2007;Manninen et al., 2010;Kulmala et al., 2012). The
classification of events utilizing PSD data that extends below 10 nm, which is a typical measurement limit for
most SMPS systems, improves the event classification and allows better identification of event days that would
otherwise be classified as undefined or non-events if only PSDs above 10 nm were used (Leino et al.,
2016;Dada et al., 2018;Brilke et al., 2020). In addition, spectrums of total particles (both neutral and charged)
are usually less ambiguous to classify than charged particle spectra (ion mode of NAIS), and the classification
of event days may be different if one only looks at these charged spectrums.

**2.5 NPF specific parameters**
*Condensation sink (CS)* is a loss term for condensable vapors used to describe their loss rate by condensation
to pre-existing aerosol surface. This term was first introduced by Kulmala et al. (2001) and it is derived based
on condensing vapor mass flux to the particles in the continuum regime and applying the transitional correction
factor ($\beta_m$) proposed by Fuchs and Sutugin (1971):

$$\text{CS} = 4\pi D \sum_i \beta_{m_i} r_i N_i = 2\pi D \sum_i \beta_{m_i} d_{p_i} N_i \,, \qquad (1)$$

where $r$, $d_p$ and $N$ are the particle radius, diameter and number concentration, respectively, in the size class $i$,
and $D$ is the diffusion coefficient of the condensing vapor calculated as recommended by Fuller et al. (1966):

$$D(H_2SO_4, air) = \frac{0.001 T^{1.75} \sqrt{\dfrac{1}{M_{H_2SO_4}} + \dfrac{1}{M_{air}}}}{P \left(\sqrt[3]{V_{H_2SO_4}} + \sqrt[3]{V_{air}}\right)^2} \,, \qquad (2)$$

where T is the temperature, M is the molar mass, P is the atmospheric pressure, and V is the diffusion volume.
Here, CS was calculated assuming that sulfuric acid is the main condensing vapor.



*Coagulation sink (CoagS)* is a loss term for freshly formed particles used to describe their loss rate by
Brownian coagulation to pre-existing aerosol surface (Kulmala et al., 2001). It is calculated as:

$$CoagS(d_p) = \sum_j K_{ij}N_j \ ,$$ (3)

where $K_{ij}$ is the Fuchs form of the Brownian coagulation coefficient (Fuchs, 1964;Seinfeld and Pandis, 2012).

*Growth rate (GR)* is the rate of change in the diameter, $d_p$, that represents the growing particle population. It
was calculated here using the NAIS negative ion mode data by first finding the time of the maximum
concentration at each diameter measured by the NAIS (maximum concentration method) (Kulmala et al., 2012)
and then deriving the growth rate as the slope of the linear fit between the diameters and time:

$$GR = \frac{dd_p}{dt} = \frac{\Delta d_p}{dt}.$$ (4)

We calculated GR at three different size ranges: between 1.5 and 3 nm ($GR_{1.5-3}$), between 3 and 7 nm ($GR_{3-7}$)
and between 7 and 20 nm ($GR_{7-20}$).

*Event start and end times* were determined based on the time evolution of the 2-4 nm particles which is the
size range suggested by Dada et al. (2018). Using this size range, we are able to capture the early stages of the
event which is unachievable if the measured PSD starts from bigger sizes. Thus, computed event start and end
234 times might differ across studies depending on the instrument used.

*Particle formation rate (J)* is the rate at which aerosol particles of certain size are formed in the atmosphere.
It quantifies the intensity of the NPF events, and it is calculated by rearranging the equation describing the
time evolution of the particle number concentration (Kulmala et al., 2012). $Dp$ in this equation refers to the
smaller limit of the size bin used in the calculation of the formation rate. We calculated $J$ at three sizes: 1.5 nm
($J_{1.5}$), 3nm ($J_3$) and 7nm ($J_7$); the upper size limits used were 3, 7 and 20 nm, respectively. GR was considered
constant within the event start and end times. Outside the event times and during non-events the GR term was
considered equal to zero.

$$J_{dp} = \frac{dN_{d_p}}{dt} + \text{CoagS } N_{d_p} + \frac{GR}{\Delta D_p}N_{d_p},$$ (5)

where the first term in represents the time evolution of particle number concentration $N_{d_p}$, the second term
represents the coagulation losses due to larger aerosol particles, and the third term represents the
condensational growth to sizes bigger than the considered size range.

### 2.6 Sulfuric acid proxy
Sulfuric acid is one of the key gas-phase compounds identified to contribute to new particle formation (e.g.
Weber et al., 1996;Sipilä et al., 2010). As direct measurements of sulfuric acid is challenging, a suite of proxies
for the sulfuric acid concentrations are derived that facilitate calculation of gas phase sulfuric acid from
ancillary observations (Dada et al., 2020;Mikkonen et al., 2011;Petäjä et al., 2009;Lu et al., 2019;Weber et al.,
1997). In this study, the sulfuric acid proxy was calculated using the new method by Dada et al. (2020) for a
rural site, which was developed based on observations from the same site of this study :

$$[H_2SO_4]_{rural} = -\frac{CS}{2 \ x \ (2 \ x \ 10^{-9})} + \left[\left(\frac{CS}{2 \ x \ (2 \ x \ 10^{-9})}\right)^2 + \frac{[SO_2]}{(2 \ x \ 10^{-9})}(9 \ x \ 10^{-9} \ x \ GlobRad)\right]^{\frac{1}{2}}$$ (6)



This proxy does not only consider the formation of $H_2SO_4$ from $SO_2$ via OH oxidation and the loss towards
pre-existing particles (condensation sink), but also includes the formation pathway via stabilized Criegee
Intermediates and loss towards atmospheric clustering starting from $H_2SO_4$ dimer formation.

**2.7 Air mass origin analysis**

Air mass origins for the entire measurement period were modeled using the Lagrangian particle dispersion
model FLEXPART (FLEXible PARTicle dispersion model), version 8.23, in a backward mode (Stohl et al.,
2005), with meteorological (0.5x0.5°, 6 h starting from midnight UTC) NCAR (ds 0.94) data as input. We
used "species" 1 (tracer), which do not include wet or dry deposition and assumes an infinite lifetime for the
particles, as the tracer released to model the retroplumes. Retroplumes replaces simple back trajectory
calculations in the interpretation of atmospheric trace substance measurements, and were traced back in time
for 5 days using CAO as the receptor site. Air masses were categorized to source regions based on the potential
emission sensitivity (PES) for the lowest 1 km above ground level (agl), following the classification method
of Pikridas et al. (2010). Seven source regions were identified similar to the ones presented by Pikridas et al.
(2018) except that in our analysis, the West Turkey sector was merged to the NW Asia sector.

**2.8 Identification of days with  high dust loading**

Measurement dates with high dust loading were identified using the VI-PM1 online method proposed by
Drinovec et al. (2020). This method couples a high-flow virtual impactor (VI) sampler, which concentrates
coarse particles, with an aerosol absorption photometer. More details about the calculations and a list of the
identified dust days can be found in Sect. 5 of SI.



## 3 Results and Discussion

In the course of identifying NPF events, the PSD spectrum is usually analyzed, mainly at sizes below 25 nm where one can detect the emergence of new aerosol particles, and then the particle growth to larger sizes is followed. Since little is known about particle number size distributions from the EMME region, we will first present the seasonal and diurnal variability of particle number concentration in different PSD modes (Sect. 3.1). Then, we will identify and characterize NPF events (Sect. 3.2). Following, we will quantify and analyze relevant parameters that describe NPF events (Sect. 3.3) and use those parameters, together with meteorological variables, to understand why and when NPF occurs at our site (Sect. 3.4). All the data in this manuscript are presented in local time (UTC+3 from 25 March 2018 to 28 October 2018, and UTC+2 during the rest of the campaign). Unless otherwise indicated, we mainly focus on daytime data having global radiation > 50 W m$^{-2}$ because it is the time period relevant for most NPF events, but we also briefly mention some night-time events. For reference, the monthly range of day hours having global radiation > 50 W m$^{-2}$ is presented in Figure S5.

### 3.1 Seasonal and diurnal variability of number concentration in different modes

Figure 2 presents the monthly percentiles boxplots (25$^{th}$, 50$^{th}$ and 75$^{th}$) and the mean averages of the cluster mode [~1–3 nm], nucleation mode [3–25 nm], Aitken mode [25–100 nm] and accumulation mode [100-1000 nm] particle number concentrations computed from daily data with global radiation > 50 W m$^{-2}$ (daytime conditions). A clear seasonal pattern is depicted which is distinct across the different particle modes. The cluster mode and nucleation mode particles had roughly a similar pattern, with the highest concentrations during the spring followed by the autumn and a clear drop during the summer. The cluster and nucleation mode concentrations can be directly linked to the NPF activity, especially in sites where direct emissions of particles having these size ranges are minimal, which is the case for our site. The Aitken mode exhibited higher concentrations during the spring months followed by a decreasing pattern, which could either suggest more growth from NPF to Aitken sizes or higher emission during spring. The accumulation mode had its maximum during the summer, except during July which did not follow the pattern of other months. Previous long-term measurements of PM$_{2.5}$ at this site have a similar pattern with higher concentrations during the warm period of the year and minimum during winter (Pikridas et al., 2018). This maximum during the summer is mainly explained by the enhanced transport of polluted air masses from the north sector, combined with the lack of precipitation and overall dry conditions during Eastern Mediterranean summer (Pikridas et al., 2018).

[Figure 2 goes here]

The diurnal variation (at radiation > 50 W m$^{-2}$) of the cluster and nucleation mode particles exhibited a clear cycle, with the highest values recorded between 9:00 and 15:00 am and the maximum at 11:00 (Figure S6 a & b). The Aitken mode had a less distinct diurnal cycle having the peak at later hours of the day, which might indicate that these particles have possibly grown from the cluster and nucleation modes (Figure S6.c). The accumulation mode, on the other hand, did not exhibit any clear diurnal cycle, which might suggest that these particles are not emitted or produced from any local source but are rather long-range transported. They can be aged primary particles, or particles originating from NPF taken place 1-3 days earlier in arriving air masses (Figure S6.d).

### 3.2 General character of the NPF events

New particle formation has been detected to occur in a variety of environments within the troposphere (Kerminen et al., 2018;Lee et al., 2019;Nieminen et al., 2018). Typically, the appearance of clusters is detected in the morning hours followed by subsequent growth. The occurrence of new particle formation events is determined by examining the time evolution of the aerosol number size distributions (e.g. Kulmala et al., 2012). An example of such events is presented in Figure S7. Throughout the one-year measurement campaign (365 days), 207 (56.7 %) days were identified as event days, 119 (32.6 %) days were identified as non-event days,





31 (8.5 %) days were undefined days and 8 (2.2 %) days did not have data mainly due to power cuts at the
station (Figure 3). The annual-median NPF frequency at CAO calculated without accounting for days with no
data amounts to 58% which belongs to the high end of the global NPF frequency distribution (Nieminen et al.,
2018) with the highest frequency being measured in South Africa (86%; Hirsikko et al., 2012). High frequency
of NPF occurrence is also observed at Saudi Arabia (73%; Hakala et al., 2019)

NPF took place throughout the year at CAO, but it had a clear seasonal pattern with a broad spring maximum,
less pronounced autumn maximum, and slightly lower frequencies during other times of the year. The months
with the highest NPF frequencies were March and April, while June and August had the lowest frequencies.
This seasonal pattern of NPF frequency is very similar to that recorded at the Finokalia atmospheric
observation station in Crete, Greece (Kalivitis et al., 2019) which is a nearby Eastern Mediterranean site having
similar synoptic conditions. Nonetheless, monthly NPF frequency at Finokalia ranged between ~17 and 42%
which is substantially lower than the range reported here for CAO (33 - 86%). The higher NPF frequency at
CAO could partially be due to the use of PSD data that starts from the ~1-nm size range, which facilitates NPF
classification especially during days when the particle growth does not pass the 10-nm size or does not continue
for several hours. We compared the NPF classification using SMPS data only and that using full PSD for time
periods when SMPS data were available, and attained 30 % less event days classified. Another factor that
could contribute to the higher NPF frequency at CAO is the surrounding forest nature which emits VOCs that
oxidize in the atmosphere and contribute to particle growth (Riipinen et al., 2011).

[Figure 3 goes here]

We further separated the NPF event days into class I or class II events, or into the so-called "bump" events
(Manninen et al., 2010). Examples of these event types are given in Figure S7. Class I events differ from class
II events not by the strength of the event but rather by the ability to calculate the particle growth rate for such
event, meaning that the growing mode diameter or concentration does not fluctuate strongly. Bump events are
NPF events where a burst of nucleation mode particles is seen but the particles do not usually grow past the
~10-nm size, and the duration of these events is typically short. The calculation of growth rates for these events
is sometimes problematic because the growth happens very fast (in less than 15 minutes) and it cannot be
captured by the time resolution of the measuring instrument. In the literature, these events have been called
"bursting events" (Dall´Osto et al., 2017), "hump events" (Vakkari et al., 2011;Yli-Juuti et al., 2009),
"suppressed events" (Chen et al., 2017), "stationary NPF events" (Größ et al., 2018) or "weak NPF events"
(Lee et al., 2020). The fraction of these events were highest during the month with the lowest NPF frequency
(mainly during summer), which could imply that during these months less amount of condensing vapors was
present to grow the particles to bigger sizes or extend the event duration (Figure 4).

[Figure 4 goes here]

The NPF events started almost always from the sub-3-nm range at CAO. The apparent growth reached a
diameter of 20 nm on 25% of event days (Figure 5a), thus it could have been difficult to identify those days if
we have relied solely on SMPS measurements which suffer from high losses and low counting statistics in the
sub-10-nm size range (Brilke et al., 2020;Kangasluoma et al., 2020;Wiedensohler et al., 2012). Additionally,
it was difficult to distinguish the growing mode at sizes above 50 nm mainly because of background aerosols
and fluctuating air masses. This implies that particles growing from NPF might have been able to grow to
bigger sizes, but their identification from the PSD spectrum was not possible. The median event duration was
~ 5.4 hours (Figure 5b). The events typically started two to four hours after sunrise and ended seven to eleven
359   hours after sunrise (Figure 5c), similar to what was observed by Dada et al. (2018).

[Figure 5 goes here]





Another feature of NPF events observed at CAO was the occurrence of two or three consecutive daytime
nucleation events (Figure 6). These multiple events occurred on ~20% of the recorded event days. Similar
observations were reported in South Africa, and they were mainly attributed to changes in air masses,
interruptions by clouds and boundary layer dynamics and its relation to the amount of vapors present (Hirsikko
et al., 2013). Salma and Németh (2019) have also showed that NPF events with broad or multiple onsets are
abundant in the urban environment of Budapest, Hungary.

We also observed events with a decreasing mode diameter, sometimes referred to as shrinkage events. These
events were mainly observed in the NAIS ion mode, while some of them were also observed in both ion and
total particle spectrum (Figure 7). These types of events have been observed in multiple environments and are
usually attributed to particle evaporation triggered by elevated temperatures, size-dependent dilution by wind-
or boundary layer development-mixing, or changes in air masses bringing small particles to the measurement
site (Alonso-Blanco et al., 2017;Backman et al., 2012;Cusack et al., 2013;Hakala et al., 2019;Kivekäs et al.,
2016;Salma et al., 2016b;Skrabalova et al., 2015;Tsagkogeorgas et al., 2017;Yao et al., 2010;Young et al.,
2013;Zhang et al., 2016;Carnerero et al., 2018).

We spotted a few events with nighttime clustering, which could reflect a chemistry that does not depend on
photo-oxidation (Figure 8). These events occurred mainly during the cold months associated with high cluster
mode concentration. Nighttime events have been observed in other Mediterranean studies as well (Carnerero
et al., 2018;Kopanakis et al., 2013;Kalivitis et al., 2012). In a boreal forest, nighttime clustering events that do
not usually grow past 5 nm have been attributed to the formation of large highly-oxygenated organic molecules
(HOM) mainly from monoterpene oxidation (Lehtipalo et al., 2011;Rose et al., 2018;Bianchi et al., 2019). In
the French Landes forest, nocturnal NPF events with clear growth up to 100 nm were attributed to monoterpene
oxidation under stratified atmospheric conditions (Kammer et al., 2018). Monoterpenes concentrations
reported at the Landes forest reached up to 25 ppb, whereas those reported in the boreal forest were below 2
384   ppb. Concurrent measurements of biogenic volatile organic compounds (BVOCs) were not available in this
study but the average concentration of monoterpenes during March 2015, which is a month with high biogenic
activity, was reported to be  $0.236 \pm 0.294$ ppb with a maximum up to 4.5 ppb (Debevec et al., 2018).

Lastly, the EMME region is characterized by a high loading of dust which contributes to around 34% (~10
$\mu g m^{-3}$) of the annual $PM_{10}$ levels (Pikridas et al., 2018). Desert dust storms occur more frequently during late
winter and spring (Achilleos et al., 2014;Pikridas et al., 2018). In this study, fifty days with high dust loading
(Table S3) were identified based on ground measurements of mineral dust concentrations (Sect. 5 of SI).
Among these dates, 37 were NPF event days, 9 were non-events, 2 were undefined and 2 had no data. Thus,
high dust loading (translated to a high condensation sink) does not seem to suppress NPF. On the contrary, Nie
et al. (2014) have suggested earlier that heavy dust plumes could be an unexpected source of nucleating and
condensable vapors via dust-induced heterogeneous photochemical processes. Further investigation is required
to prove this suggestion for our site.

### 3.3 NPF specific parameters
In this section we analyze three parameters related to NPF: particle formation rates ($J$), particle growth rates
(GR) and condensation sink (CS). $J$ and GR describe the strength of NPF, while CS is a measure of the sink
for both vapors contributing to NPF and small growing clusters. Taken together, these three quantities
eventually determine how efficiently NPF contributes e.g. to atmospheric CCN production.

*Particle formation rates:* The particle formation rates for 1.5, 3 and 7 nm particles ($J_{1.5}$, $J_3$ and $J_7$, respectively)
were calculated and they are presented in Table 1(daily data) and Table S4 (hourly data). $J_{1.5}$ was the highest
during the spring: March had the highest median $J_{1.5}$ while April had more events with extreme $J_{1.5}$ values as
expressed by the higher mean. We were not able to calculate formation rates later than the $2^{nd}$ of November

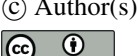



because SMPS measurements were not available, but the pattern of $J_{1.5}$ seems to exhibit another peak during
the autumn, which is similar to the seasonal pattern of the NPF frequency and cluster mode concentration. In
contrast, $J_3$ and $J_7$ did not exhibit a clear seasonality, but their values were in general higher during the spring.
The diurnal cycle for the formation rates was more pronounced during the Class I events than during the Class
II or bump events, and the peak median hourly value was highest during Class I events (Figure 9). The median
peak of $J_{1.5}$ and $J_3$ during the class I events and bump events occurred between 11:00 and 12:00, whereas for
$J_7$ the peak occurred between 12:00 and 14:00. For the Class II events the corresponding peaks occurred about
1 hour later. To place the formation rates in global perspective, we compare $J_3$ from this study to other studies
(Table 2), because it is the most commonly reported value in literature. The studies in Table 2 were selected
on the basis of having one year or more of measurement data. $J_3$ determined in this study was up to a one order
of magnitude higher than that measured at semi-pristine rural areas (Värriö, Hyytiälä, and Tomsk), lower than
that measured in a megacity (Beijing) and comparable to values reported at urban and rural sites affected by
urban pollution (Budapest, Vavihill, and Po Valley).

*Growth rates:* We report size-segregated growth rates between 1.5 and 3 nm (GR $_{1.5-3}$), between 3 and 7 nm
(GR $_{3-7}$), and between 7 and 20 nm (GR $_{7-20}$) as recommended by Kulmala et al. (2012) (Table 3). The median
growth rates in these size ranges were 2.0, 4.8, and 7.4 nm hr$^{-1}$, respectively. These GRs are higher than those
reported for a rural boreal environment (1.9, 3.8 and 4.3 nm hr$^{-1}$, respectively) (Yli-Juuti et al., 2011), but
within the range of GRs reported for 12 European sites (Manninen et al., 2010 cf. Figure S8). The increase in
the growth rate with an increasing particle size is a typical feature in the sub-20 nm size range because
condensational growth is more favorable as the particle size increases and the Kelvin effect decreases
(Manninen et al., 2010). Additionally, we calculated the growth rates between 3 and 25 nm (GR $_{3-25}$) for the
purpose of making comparison with additional studies. Similar to $J_3$, the median GR $_{3-25}$ calculated here (6.3
427 nm hr$^{-1}$) was comparable to the range reported for urban or rural sites with urban influence (4.7 – 7.7 nm hr$^{-1}$),
whereas rural sites usually have a GR $_{3-25}$ below 4 nm hr$^{-1}$. In regard to the seasonal variability, we did not find
a clear pattern in GR $_{7-20}$ or GR $_{3-25}$. In contrast, GR $_{1.5-3}$ and GR $_{3-7}$ were generally higher during the summer
430 months, which could be associated to the higher fraction of bump events with respect to other event types. As
discussed in Sect. 3.3, these bump events are characterized by a burst of particles within a short period of time,
which would translate to higher growth rates.

*Condensation sink:* The median of CS at CAO, for the periods that SMPS measurements were available, was
$7.9 \times 10^{-3}$ s$^{-1}$ (25th - 75th percentiles = $5.2 \times 10^{-3}$ – $13.9 \times 10^{-3}$) while the mean was $10.7 \times 10^{-3}$ s$^{-1}$ $\pm 8.2 \times 10^{-3}$ s$^{-1}$
(computed from daily median values). These values lie within the range of coastal (Kalivitis et al., 2019) and
urban environments (Salma et al., 2016a;Jun et al., 2014). They are higher than the values reported for forests
and semi-pristine environments (Dal Maso et al., 2002;Dada et al., 2017), and lower than the values reported
for highly polluted cities (Wu et al., 2007). The average diurnal cycle of the size segregated CS for the whole
measurement period shows that particles above 50 nm were the main contributors to the CS, even though
particles down ~3 nm could also exhibit a high CS (Figure S9). Thus, nucleating aerosols can largely contribute
to the available aerosol surface area. The seasonal variation of the CS followed the seasonal pattern of the
accumulation mode particles with highest values calculated for winter and spring, and lowest for summer and
autumn. This pattern does not follow the pattern reported from measurements carried out at Finokalia (Kalivitis
et al., 2019), where the highest values were calculated during the summer and autumn (Table S5). More
analysis about the seasonal variation of CS, especially with respect to NPF events, will be presented in the
Sect. 3.4.

**3.4 The driving atmospheric parameters of the NPF events**
To explain the occurrence of NPF at CAO, we investigated the effect of the following variables: CS,
meteorological conditions (temperature, solar radiation, pressure, relative humidity, wind speed and wind



direction), trace gas concentrations (NO$_x$, SO$_2$, CO and O$_3$), air mass origin and sulfuric acid – formation rate
relationship.

The NPF frequency typically decreases with an increasing CS (Pikridas et al., 2012;Salma et al., 2016a;Dada
et al., 2017;Dai et al., 2017;Hakala et al., 2019;Hussein et al., 2020). However, NPF has been observed in
polluted environments at exceptionally high values of CS, indicating that inefficient cluster scavenging or
enhanced cluster growth or a combination of both drives NPF regardless of the high load of pre-existing
particles (Yao et al., 2018;Kulmala et al., 2017). In our study, we did not find a clear relation between CS and
the monthly NPF occurrence, and NPF did not necessarily occur at low values of CS (Figure 10). To further
explore the effect of CS on NPF, we checked out whether the NPF event days had lower CS before the onset
of nucleation (period from midnight to morning) in comparison to non-event days, but we did not find any
apparent association (Figure S10).

Next, we inspected the effect of meteorological variables (Figure 11) on the occurrence of NPF. By considering
the data from all the months together, NPF events took place over a wide range of meteorological conditions.
Higher temperatures seemed to be favorable for intra-monthly NPF occurrence, however the higher
temperatures from June to September did not coincide with higher NPF frequencies (Figure 11a). The effect
of temperature on NPF has been studied extensively in chamber experiments, with a general consensus that
lower temperatures favor nucleation at the kinetic regime and thus enhance NPF in inorganic systems like the
sulfuric acid-ammonia system (Lee et al., 2019). However, in organic systems where highly oxygenated
organic compounds (HOM) are the main NPF species, temperature plays a double role. On the one hand, the
Gibbs free-energy barrier is reduced at lower temperatures, favoring the condensation of less oxidized vapors
that would not condense at higher temperatures. On the other hand, lower temperatures lead to decreased auto-
oxidation reaction rates and reduced yields of HOM. Recent studies have shown that the former effect
compensates for the latter effect, having an overall increase in nucleation and growth rates at lower
temperatures (Stolzenburg et al., 2018;Simon et al., 2020;Ye et al., 2019). While these mechanisms are clear
in chamber studies, the situation becomes more complicated in the atmosphere because of the complexity of
the atmosphere-biosphere system having simultaneous temperature-dependent processes that can enhance or
suppress NPF, making current atmospheric observations inconsistent on the role of temperature on NPF
(Kerminen et al., 2018). Solar radiation is regarded as one of the most important factors affecting NPF (Jokinen
et al., 2017). Its intensity is relatively high in Cyprus all year round. Intra-monthly, NPF events occurred at
higher global radiation during the winter and autumn month, whereas in spring (except April) and summer
480  months, radiation did not seem to be a limiting factor for NPF (Figure 11b). Inter-monthly, the month with the
highest solar radiation did not coincide with the highest occurrence of NPF. Regarding ambient relative
humidity (RH), NPF events tend to occur at lower RH in both clean and polluted environments (Kerminen et
al., 2018). However, high RH values do not necessarily suppress NPF (Salma and Németh, 2019), which agrees
with our observations (Figure 11c). In terms of the surface air pressure, intra-monthly NPF was on average
observed on days with higher pressures and the inter-monthly NPF occurrence was the lowest during the month
with the lowest surface pressure (Figure 11d). NPF occurred largely at lower wind speeds and local north-
easterly winds, which is the direction where the main agglomerations and livestock farming lands are situated
(Figure 11e, 11f & S11).

The EMME region is among the regions with the highest background of trace gases and aerosols concentrations
in the Northern Hemisphere (Lelieveld et al., 2002). Here, we investigate the relation between trace gases and
the occurrence of NPF (Figure 12). The intra-monthly SO$_2$ concentration was, on average, higher during NPF
event days in comparison to non-event days during most of the months. The inter-monthly SO$_2$ concentrations
during the highest NPF occurrence (Mar-May) were similar to the months with the lowest NPF occurrence
(Jun-Aug). This indicates that SO$_2$ and thus sulfuric acid (as will be shown subsequently) cannot explain the
seasonal pattern of NPF. The ozone (O$_3$) concentration is particularly high in Cyprus and is mainly influenced



by regional and transported ozone, while local precursor emissions play a minor role in ozone formation
(Kleanthous et al., 2014). Intra-monthly, the $O_3$ concentration was sometimes lower, similar or higher during
NPF event days compared with non-event days, with no clear seasonality. One notable remark is that in April,
NPF events took place at much higher $O_3$ concentrations than what was observed on non-event days. This
could imply that higher oxidative capacity was driving NPF during April. April also had the most notable
differences in global radiation and RH between NPF event and non-event days. Analogous to $SO_2$, $O_3$ cannot
explain the seasonal pattern of NPF. CO levels are generally high over the Mediterranean (in comparison to
the Pacific), with emission sources being typically from western and eastern Europe, having lower contribution
from the regions surrounding the Mediterranean (Lelieveld et al., 2002). The CO - NPF relationship at CAO
did not have a distinct character, however CO concentrations were slightly lower during the summer months.
Regarding $NO_x$, NPF event days had on average higher $NO_x$ concentrations within the boundaries of each
507  month, except in April. More notably, $NO_x$ had lower concentrations during the months with lower NPF
frequencies, which might indicate the role of associated anthropogenic organic vapors in triggering NPF at our
site.

We examined the effect of air mass origin arriving at CAO at 8:00 a.m. during event and non-event days from
six source regions: local, N. Africa, marine, Europe, Asia, NW. Asia, and SW. Asia (Figure 13). The last two
source regions represent the geographic location with respect to Cyprus location. An obvious feature that pops
out is that the month with the highest NPF frequency had air masses originating from south-west Asia (the
Middle East), whereas during the month with the lowest NPF frequency air masses did not originate from that
direction. This pattern might suggest that chemical compounds important for nucleation and subsequent growth
are transported to CAO from the Middle East. Between the end of spring and late September, which are the
517  months with the lowest NPF frequency, the circulation over the eastern Mediterranean is characterized by
persistent northerly winds called the Etesians (Tyrlis and Lelieveld, 2013). The NPF events during this period,
as shown in Sect. 3.2, were weak or did not lead to particle growth into large sizes in comparison to the rest of
the year. The Etesian circulation flow is caused by a sharp surface pressure difference between the westerly
Azorean high-pressure regime and the Asian monsoon low-pressure regime. While the Etesians block the
north-ward transport of desert dust, they trigger high sea levels, prevent rain over the region, and enhance
marine inversions (Ulbrich et al., 2012). They favor the transport of air pollutants from Central/Eastern Europe
and west Turkey and, together with enhanced photochemical conditions and low precipitation, contribute to
high $O_3$ (Solomou et al., 2018) and particulate matter (PM) levels (Pikridas et al., 2018). The increase in PM
levels during these months could be a limiting factor for NPF. Indeed, accumulation mode particles and thus
CS were the highest during the summer (except for July). We did not have particle size distribution
measurements above 700 nm, therefore bigger particles could be additionally contributing to the CS. However,
from mass concentration point of view, $PM_{2.5}$ and $PM_{10}$ did not show a pattern that would support this
hypothesis (Figure S12 & S13). Additionally, while the south-west Asia sector might be important for NPF, it
did not exhibit a clear pattern during the month with the highest NPF frequency. In fact, in April most of the
air masses originated from north-west Asia. This sector appears to be also important for NPF during Jun, July
and November, whereas the other sectors did not exhibit any notable pattern.

Last, we investigated the relationship between sulfuric acid and formation rates. While sulfuric acid ($H_2SO_4$)
is the main nucleating species in the atmosphere, it is well known that $H_2SO_4$ binary nucleation with water
requires high $H_2SO_4$ vapor concentrations that are not relevant within the lower parts of the troposphere
(Wyslouzil et al., 1991). Additional species are required to stabilize $H_2SO_4$ clusters, such as ammonia, amines
or ions, while some other compounds can nucleate on their own in atmospherically relevant conditions,
including iodine oxides and highly oxygenated organic compounds (HOM) from biogenic precursors (Lee et
al., 2019 and references therein). In this study, the hourly $H_2SO_4$ proxy concentrations ranged between $3 \times 10^5$
and $1 \times 10^7$ cm$^{-3}$ which are typical values for $H_2SO_4$ in the troposphere. The relationship between particle

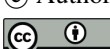



formation rates ($J_{1.5}$) and H$_2$SO$_4$ proxy concentration varied across the month of the year (Figure 14). Lower
concentrations of H$_2$SO$_4$ were required during winter and spring to achieve the same formation rates as in the
other seasons. A possible explanation to this behavior is that in the first case, stabilizing compounds are
abundant in the atmosphere and thus less H$_2$SO$_4$ is required for the formation of particles. A similar hypothesis
was tested by Pikridas et al. (2012) by using the accumulation mode particle acidity as an indirect measure of
the availability or lack of ammonia or any other basic species in the gas phase. The authors concluded that
excess base is not available during the summer to participate in the nucleation process. In our case, however,
the formation rate versus H$_2$SO$_4$ relationship is closer to those derived for the H$_2$SO$_4$-DMA-NH$_3$-H$_2$O system
than those for the H$_2$SO$_4$-NH$_3$-H$_2$O system. In fact, the ternary nucleation of H$_2$SO$_4$ -NH$_3$-H$_2$O is unlikely to
be important at ground level either because of too low concentrations or too high temperatures (Kürten et al.,
2018). This suggests that, in our case, the missing stabilizing base is probably not ammonia. The distinct air
mass origin during the summer could explain the decrease in the concentrations of the stabilizing base.
Otherwise, the high temperature during the summer could be the factor that disfavors the occurrence of NPF.
Most certainly, NPF at CAO seem to be influenced by several factors and chemical constituents. This has been
also indicated by Debevec et al. (2018) whom observed four types on nucleation events, within one month of
measurements, having: 1) predominant anthropogenic influence,  2) predominant biogenic influence, 3) mixed
anthropogenic - biogenic influence, and 4) a marine influence with low concentrations of anthropogenic and
biogenic tracers. Therefore, to reveal the main mechanisms of NPF, long-term measurements of nucleating
clusters and organic precursors using state-of-the-art online mass spectrometry techniques are essential.

**4    Conclusion**

Recent studies have pointed out that NPF is important in the EMME region (Brilke et al., 2020;Debevec et al.,
2018;Hakala et al., 2019;Hussein et al., 2020;Kalivitis et al., 2019;Kalkavouras et al., 2019). Brilke et al.
(2020) studied NPF in a coastal site in Cyprus with strong local pollution during 2017, while Debevec et al.
(2018) characterized NPF at the same site of this study during 2015. While both studies were limited to one
566    month of observations, we disclosed here the first long-term (one year) characterization of NPF at a
background site in Cyprus. We presented the general and seasonal characteristics of PSD and NPF then we
explored the factors that affect NPF.
Our analysis of NPF intra-monthly variability showed that on average, NPF events occurred at higher
temperatures, lower RH and higher global radiation, except during the months of August, September and
December. To the contrary, lower pressure conditions, higher wind speeds and local northwest wind directions
seemed to be more favorable for non-events. The frequency of NPF was higher than that reported at a similar
Eastern Mediterranean island site using a slightly limited measurement setup than the one applied here. This
demonstrates the importance of comprehensive measurements using instruments that can measure down to
cluster sizes. NPF occurred all year round, with higher frequencies during the spring and autumn and a
minimum frequency during the summer. The particles did not grow significantly after nucleation during the
577    months with the lowest NPF frequencies. These months were also characterized by lower NO$_x$ concentrations,
an indication of lower anthropogenic influence, and distinct air mass origin profiles from the rest of the year.
Condensation sink, calculated based on PSD up to 700 nm, had no clear relationship with NPF, but it was
slightly higher during some summer month. Additionally, sulfuric acid was not the limiting factor for NPF
occurrence as its estimated concentration was mostly high during the summer, up to 1e$^7$ molecules.cm$^{-3}$.The
relationship between particle formation rates and sulfuric acid proxy exhibited different slopes between the
583    months with the highest and lowest NPF frequency, suggesting that nucleation might have proceeded with
varying temperatures or at different concentrations of stabilizing compounds and other aerosol precursors not
measured in this study.
The analysis presented in this study is a step forward towards understanding the mechanisms of NPF
mechanism in the EMME region. Future studies require long-term measurements of vapors that participate in
NPF and subsequent growth. These could include, for example, ultra-low volatility organic compounds





(ULVOC), extremely low-volatility organic compounds (ELVOC), low-volatility organic compounds
(LVOC), ammonia, amines and iodic species. Further, to understand the ubiquity of the effect of large particles
which could inhibit NPF during certain episodes but enhance NPF during episodes with high mineral dust
loadings, extended PSD measurements up to coarse particles, preferably coupled with chemical speciation, are
important. On a larger scale, long-term measurements of CCN particles are necessary to quantify the
contribution of NPF to the CCN budget. These measurements would preferably take place not only in Cyprus
but also in different location in the Middle East and North Africa.



**Data availability**
Data used in this study are available from the corresponding author upon request (rima.baalbaki@helsinki.fi)

**Author Contributions**
The study was conceived by MK and JS. RB, TJ, TL, KN and MP prepared and installed the instruments. RB,
MP, KN, AM, EB, AC, and FU performed the regular maintenance for the instruments. RB performed the data
analysis and wrote the manuscript. LD provided support in data analysis. LA provided support in instrument
troubleshooting and nCNC inversion. MP and EB performed the sector analysis. SB performed the kappa
measurements and provided support on hygroscopisity calculations. MP and JS provided the SMPS data. CS
provided the meteorological and trace gas data. RB, MP, TJ, LD, SB, KL, JK, GB, TP, VMK, and MK
participated in the scientific discussion and reviewed the manuscript.

**Competing interests**
The authors declare no conflict of interest.

**Financial support**
This publication has been produced within the framework of the EMME-CARE project which received funding
from the European Union's Horizon 2020 Research and Innovation Programme, under Grant Agreement No.
856612 and from the Cyprus Government. This work has received additional funding from the European
Research Council (ERC) under the European Union's Horizon 2020 research and innovation programme
(ERC, Project No.742206 "ATM-GTP"). The sole responsibility of this publication lies with the author. The
European Union is not responsible for any use that may be made of the information contained therein.
Additional support was received from the Academy of Finland (Grant Agreement No. 307331 and 316114),
the European Regional Development Fund and the Republic of Cyprus through the Research and Innovation
Foundation (Project: INTEGRATED/0916/0016). JK acknowledges support from Academy of Finland
(project 1325656) and University of Helsinki 3 year grant 75284132. TJ acknowledges support from Academy
of Finland (project 334514).

**Acknowledgments**
The authors thank Hanna Manninen, Kaspar Dällenbach, Jenni Kontkanen, Runlong Cai and Dominik
Stolzenburg for fruitful scientific discussions. Frans Korhonen, Pekka Rantala, Pasi Aalto, Erki Siivola, Sander
Mirme, Joonas Vanhanen, and Aki Halonen, are kindly acknowledged for their continuous and indispensable
support.

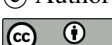



Table 1. Monthly values of observed formation rates (cm$^{-3}$ s$^{-1}$) during NPF events (calculated within the event duration using daily data). The maximum values and minimum values of the mean and median are highlighted.

| | | Jan | Feb | Mar | Apr | May | Jun | Jul | Aug | Sep | Oct | Nov | Dec | All |
|---|---|---|---|---|---|---|---|---|---|---|---|---|---|---|
| **J$_{1.5}$** **(cm$^{-3}$.s$^{-1}$)** | Mean | 6.43 | 17.81 | 19.98 | **39.71** | 6.78 | na | 4.50 | 10.36 | 10.29 | 5.61 | 8.15 | na | 16.18 |
| | SD | 2.94 | 24.77 | 14.84 | 93.97 | 6.39 | na | 3.00 | 10.60 | 10.77 | 6.28 | na | na | 41.82 |
| | 25$^{th}$ | 4.28 | 6.18 | 6.13 | 5.82 | 3.35 | na | 1.99 | 3.26 | 1.01 | 0.95 | na | na | 3.74 |
| | Median | 6.83 | 10.19 | **18.52** | 12.46 | 4.46 | na | 4.07 | 6.97 | 8.99 | 3.53 | 8.15 | na | 7.87 |
| | 75$^{th}$ | 8.39 | 16.96 | 29.71 | 34.11 | 6.92 | na | 7.32 | 13.84 | 14.23 | 9.09 | na | na | 16.01 |
| | 90$^{th}$ | 10.18 | 34.28 | 42.52 | 69.59 | 17.61 | na | 8.30 | 27.07 | 23.60 | 12.34 | na | na | 35.26 |
| | N | 5 | 16 | 21 | 23 | 17 | na | 9 | 10 | 13 | 18 | 1 | na | 133 |
| **J$_3$** **(cm$^{-3}$.s$^{-1}$)** | Mean | 1.57 | 4.68 | 5.65 | **8.77** | 3.00 | 3.48 | 5.03 | 3.85 | 5.53 | 3.00 | 2.42 | na | 4.97 |
| | SD | 0.91 | 3.44 | 4.65 | 15.74 | 2.48 | 2.60 | 6.57 | 4.19 | 5.58 | 5.16 | na | na | 7.79 |
| | 25$^{th}$ | 0.81 | 2.31 | 1.96 | 1.98 | 0.88 | 0.98 | 0.82 | 0.93 | 0.38 | 0.31 | na | na | 0.99 |
| | Median | 1.16 | 4.16 | 3.81 | **4.14** | 2.04 | 3.32 | 2.23 | 1.72 | 4.93 | 1.11 | 2.42 | na | 2.69 |
| | 75$^{th}$ | 2.54 | 5.93 | 7.72 | 8.88 | 4.45 | 6.23 | 5.88 | 7.83 | 10.06 | 2.42 | na | na | 6.42 |
| | 90$^{th}$ | 2.56 | 9.30 | 13.00 | 11.44 | 6.46 | 6.57 | 18.15 | 10.65 | 12.46 | 8.98 | na | na | 11.17 |
| | N | 5 | 16 | 21 | 26 | 20 | 6 | 13 | 11 | 14 | 19 | 1 | na | 152 |
| **J$_7$** **(cm$^{-3}$.s$^{-1}$)** | Mean | 0.58 | **1.53** | 1.17 | 1.51 | 1.30 | 0.45 | 1.59 | 0.49 | 1.32 | 0.71 | 1.15 | na | 1.18 |
| | SD | 0.51 | 1.17 | 1.09 | 1.91 | 1.44 | 0.34 | 2.20 | 0.72 | 1.75 | 0.59 | na | na | 1.43 |
| | 25$^{th}$ | 0.19 | 0.35 | 0.26 | 0.35 | 0.20 | 0.26 | 0.09 | 0.09 | 0.08 | 0.17 | na | na | 0.22 |
| | Median | 0.29 | **1.57** | 0.78 | 0.83 | 0.57 | 0.36 | 0.38 | 0.35 | 0.59 | 0.39 | 1.15 | na | 0.61 |
| | 75$^{th}$ | 1.08 | 2.13 | 1.70 | 2.04 | 2.41 | 0.55 | 2.71 | 0.56 | 1.69 | 1.37 | na | na | 1.57 |
| | 90$^{th}$ | 1.25 | 2.72 | 2.82 | 3.33 | 3.72 | 1.02 | 5.52 | 1.41 | 3.48 | 1.50 | na | na | 3.04 |
| | N | 5 | 16 | 21 | 26 | 20 | 6 | 13 | 11 | 14 | 19 | 1 | na | 152 |





Table 2. Comparison of formation rates and growth rates from long-term measurements at various sites including this study

| Site | Type | Period | $J_3$ | | | $GR_{3-25}$ | | | Reference |
|---|---|---|---|---|---|---|---|---|---|
| | | | Mean±SD | Median | Min-max | Mean±SD | Median | Min-max | |
| CAO, Cyprus | Rural | 2018 | 4.97±7.79 | 2.69 | 0.04-78.85 | 7.94±5.35 | 6.25 | 2.31-31.9 | This study |
| Vavihill, Sweden | Rural | 2001-2004 | 4.3 | 1.89 | | 2.5 | 2.1 | | (Kristensson et al., 2008) |
| Hyytiälä, Finland | Rural | 1996-2004 | 0.8 | 0.6 | 0.06–5.0 | 3.0 | 2.5 | 0.5-15.1 | (Dal Maso et al., 2007) |
| Värriö, Finland | Rural | 1996-2004 | 0.2 | 0.1 | | 2.7 | 2.4 | 0.8–9.7 | (Dal Maso et al., 2007) |
| Tomsk, Russia | Rural | 2005-2006 | 0.4 | 0.4 | 0.04-1.1 | 5.5 | 3.5 | 2.6-23 | (Dal Maso et al., 2008) |
| Beijing, China | Urban clean Urban polluted | 2004-2005 | 22.3±15.1 16.2±12 | 18.7 9.9 | 4.4-81.4 3.3-51.7 | 1.8±2 4.4±3.2 | 1.1 3.9 | 0.1-7.9 0.3-11.2 | (Wu et al., 2007) |
| St. Louis, U.S. | Urban | 2001-2003 | 17±20 | 9 | | 5.9±4.7 | 4.7 | | (Qian et al., 2007) |
| Po Valley, Italy | Rural w. urban influence | 2002-2005 | 5.89 | 3.31 | 0.24-36.89 | 6.82 | 6.45 | 2.9-22.9 | (Hamed et al., 2007) |
| Budapest, Hungary | Urban | 2008-2009 | 4.2±2.5 [a] | 4.2 [a] | 1.65-12.5 [a] | 7.7±2.4 [a] | 7.7 [a] | 2-17.3 [a] | (Salma et al., 2011) |
| Helsinki, Finland | Urban | 1997-2006 | 2 [b] | 1.09 [b] | 0.63-2.87 [b] | 3.39 [b] | 2.85 [b] | 2.21-6.55 [b] | (Hussein et al., 2008) |
| Zeppelin, Norway | Arctic | 2016-2018 | | | 0.02-1.62 | | | 0.35-7.55 | (Lee et al., 2020) |
| Finokalia, Greece | Remote coastal | 2008-2009 | | | | 7.5±5.8 [c] | | | (Pikridas et al., 2012) |

[a] data represent 6-25 nm range

[b] calculated from monthly values

[c] GR for 7-20nm



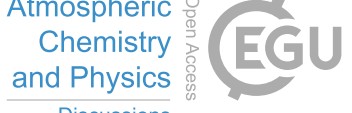

Table 3. Monthly values of the observed growth rates (nm.h$^{-1}$) computed mainly during class I events or when possible.

|  |  | Jan | Feb | Mar | Apr | May | Jun | Jul | Aug | Sep | Oct | Nov | Dec | All |
|---|---|---|---|---|---|---|---|---|---|---|---|---|---|---|
| **GR** (1.5 – 3 nm) | **Mean** | 1.17 | 1.74 | 3.19 | 2.80 | 3.55 | 1.70 | 5.22 | 4.20 | 4.26 | 1.80 | 3.94 | 5.70 | 3.01 |
|  | **SD** | 0.87 | 0.96 | 2.13 | 1.71 | 2.54 | na | 5.37 | 3.74 | 3.47 | 1.05 | 3.04 | 2.07 | 2.44 |
|  | **25th** | 0.68 | 1.00 | 1.42 | 1.57 | 2.02 | na | 1.43 | 1.93 | 1.79 | 1.23 | 1.17 | 4.52 | 1.27 |
|  | **Median** | 1.08 | 1.05 | 2.42 | 2.12 | 3.05 | 1.70 | 5.22 | 2.66 | 2.09 | 1.62 | 3.26 | 5.43 | 2.04 |
|  | **75th** | 1.87 | 2.46 | 4.46 | 3.95 | 5.09 | na | 9.02 | 6.47 | 7.83 | 1.80 | 7.10 | 6.43 | 4.25 |
|  | **90th** | 2.37 | 3.11 | 6.52 | 5.70 | 7.08 | na | 9.02 | 9.73 | 8.94 | 3.46 | 7.17 | 9.14 | 7.10 |
|  | **N** | 9 | 7 | 10 | 10 | 4 | 1 | 2 | 4 | 7 | 9 | 5 | 5 | 73 |
| **GR** (3-7 nm) | **Mean** | 4.40 | 7.13 | 6.76 | 4.52 | 5.23 | 10.88 | 7.46 | 4.11 | 10.56 | 5.28 | 6.18 | 5.68 | 6.21 |
|  | **SD** | 3.44 | 6.19 | 3.75 | 1.94 | 1.17 | na | 3.01 | 2.16 | 8.82 | 3.26 | 5.15 | 3.60 | 4.58 |
|  | **25th** | 2.45 | 3.48 | 3.78 | 3.23 | 4.66 | na | 5.64 | 2.30 | 2.31 | 3.55 | 2.80 | 3.01 | 3.25 |
|  | **Median** | 4.03 | 4.61 | 5.92 | 4.22 | 5.71 | 10.88 | 7.53 | 3.90 | 6.81 | 4.80 | 3.21 | 4.64 | 4.84 |
|  | **75th** | 6.73 | 6.79 | 8.63 | 5.35 | 5.89 | na | 8.89 | 5.93 | 17.19 | 5.71 | 8.66 | 7.67 | 6.83 |
|  | **90th** | 9.46 | 18.26 | 13.43 | 6.25 | 6.24 | na | 11.96 | 6.56 | 23.10 | 9.17 | 15.04 | 11.57 | 12.94 |
|  | **N** | 9 | 10 | 16 | 14 | 6 | 1 | 5 | 4 | 9 | 12 | 9 | 5 | 100 |
| **GR** (7 –20 nm) | **Mean** | 7.11 | 21.13 | 8.67 | 7.74 | 11.20 | 5.94 | 20.32 | 13.03 | 10.77 | 7.24 | 7.00 | 8.06 | 10.60 |
|  | **SD** | 8.30 | 16.34 | 5.28 | 3.39 | 8.02 | na | 15.62 | 14.36 | 5.29 | 2.72 | 5.89 | 4.08 | 9.45 |
|  | **25th** | 2.35 | 7.39 | 5.01 | 6.58 | 4.94 | na | 7.73 | 4.96 | 7.06 | 5.76 | 2.70 | 4.38 | 5.40 |
|  | **Median** | 5.14 | 22.10 | 6.39 | 7.09 | 8.25 | 5.94 | 16.64 | 7.17 | 10.45 | 6.95 | 5.40 | 8.11 | 7.40 |
|  | **75th** | 7.88 | 26.21 | 10.48 | 10.70 | 17.26 | na | 25.54 | 21.10 | 15.11 | 9.04 | 8.65 | 11.65 | 12.58 |
|  | **90th** | 21.05 | 45.83 | 19.40 | 12.77 | 23.54 | na | 45.95 | 34.38 | 17.85 | 10.64 | 16.30 | 12.81 | 22.44 |
|  | **N** | 8 | 11 | 15 | 14 | 8 | 1 | 6 | 4 | 8 | 12 | 9 | 6 | 102 |
| **GR** (3 –25 nm) | **Mean** | 5.98 | 13.96 | 8.88 | 6.21 | 7.80 | 5.66 | 12.11 | 5.65 | 7.37 | 5.89 | 5.69 | 7.63 | 7.94 |
|  | **SD** | 1.49 | 10.74 | 5.27 | 2.43 | 3.88 | na | 4.66 | 1.90 | 3.69 | 2.29 | 2.97 | 2.19 | 5.35 |
|  | **25th** | 5.14 | 5.10 | 5.37 | 4.67 | 5.06 | na | 9.15 | 4.05 | 4.51 | 4.32 | 3.03 | 5.47 | 4.68 |
|  | **Median** | 6.08 | 9.45 | 7.27 | 5.67 | 6.57 | 5.66 | 9.99 | 5.43 | 6.27 | 5.12 | 5.51 | 8.74 | 6.25 |
|  | **75th** | 7.05 | 23.91 | 10.38 | 6.62 | 10.94 | na | 16.84 | 7.25 | 9.95 | 6.78 | 8.93 | 9.38 | 9.31 |
|  | **90th** | 7.63 | 30.81 | 18.28 | 10.63 | 13.85 | na | 17.24 | 7.78 | 13.09 | 9.78 | 9.60 | 9.55 | 13.96 |
|  | **N** | 6 | 11 | 13 | 14 | 7 | 1 | 5 | 4 | 8 | 12 | 9 | 5 | 95 |



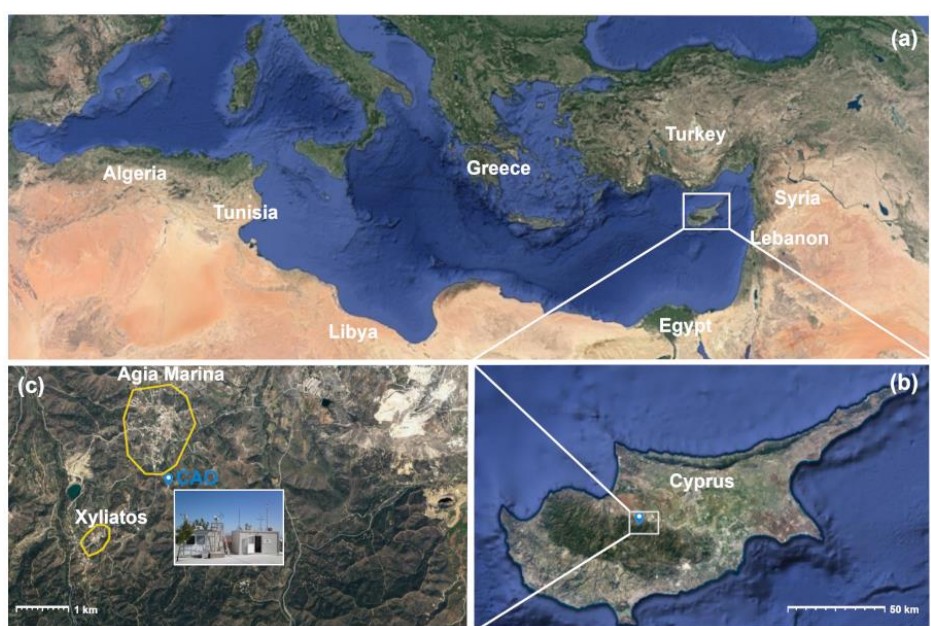

Figure 1. Maps of the Mediterranean region and Cyprus. (a) Location of Cyprus in the Mediterranean region. (b) Location of the measurement site (CAO) in Cyprus. (c) Location of the measurement site (CAO) pointed by the blue location marker with respect to the villages of Agia Marina and Xyliatos. The geographic border of the villages is marked by the yellow enclosure. The maps were retrieved from Google (©2020 Google, TerraMetrics).



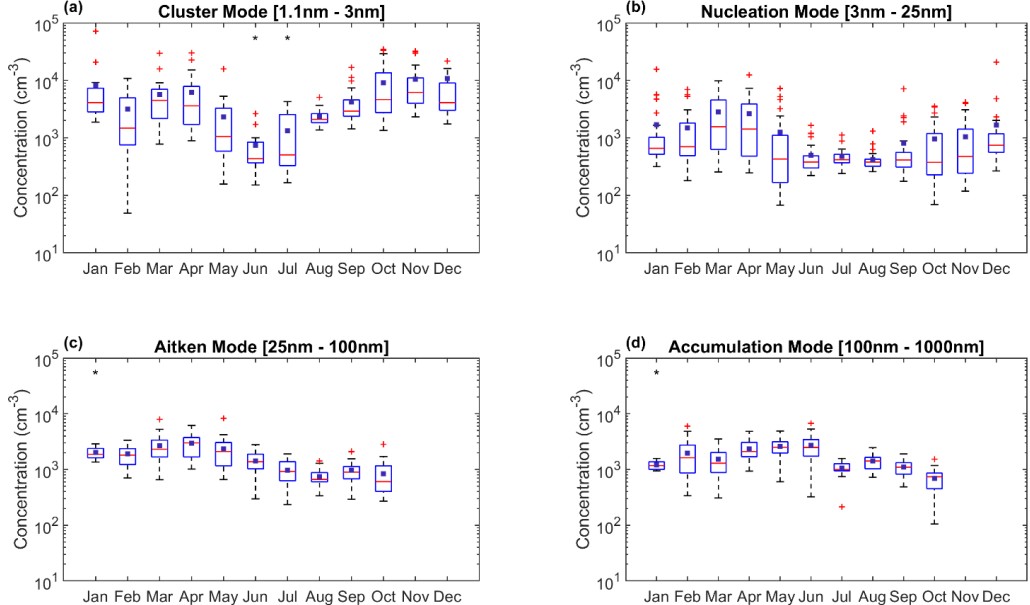

Figure 2. Monthly variation (at radiation >50 W. m$^{-2}$) of particle number concentration of (a) cluster mode, (b) nucleation mode, (c) Aitken mode, and (d) accumulation mode presented by box plots. The central red marks indicate the median, the blue small boxes indicate the mean, the bottom and top edges of the big box indicate the 25$^{th}$ and 75$^{th}$ percentiles, respectively. The whiskers extend to the most extreme data points not considered outliers, and the outliers are plotted individually using the '+' symbol. Data presented have daily time resolution. Month designated with '*' symbol have less than 20 days of data. Note that SMPS measurements were not available on November and December.

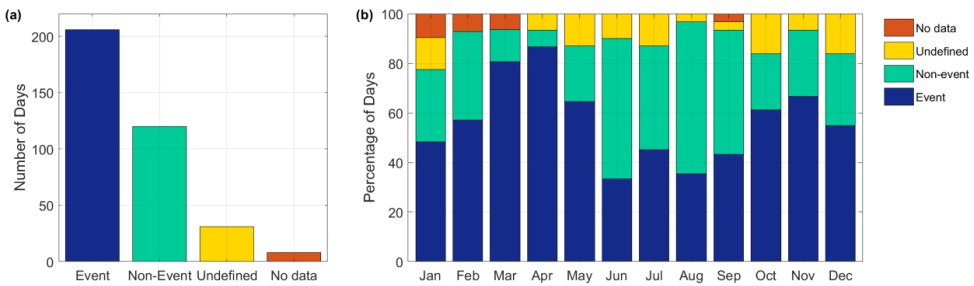

Figure 3. Classification of NPF events presented (a) annually and (b) seasonally.





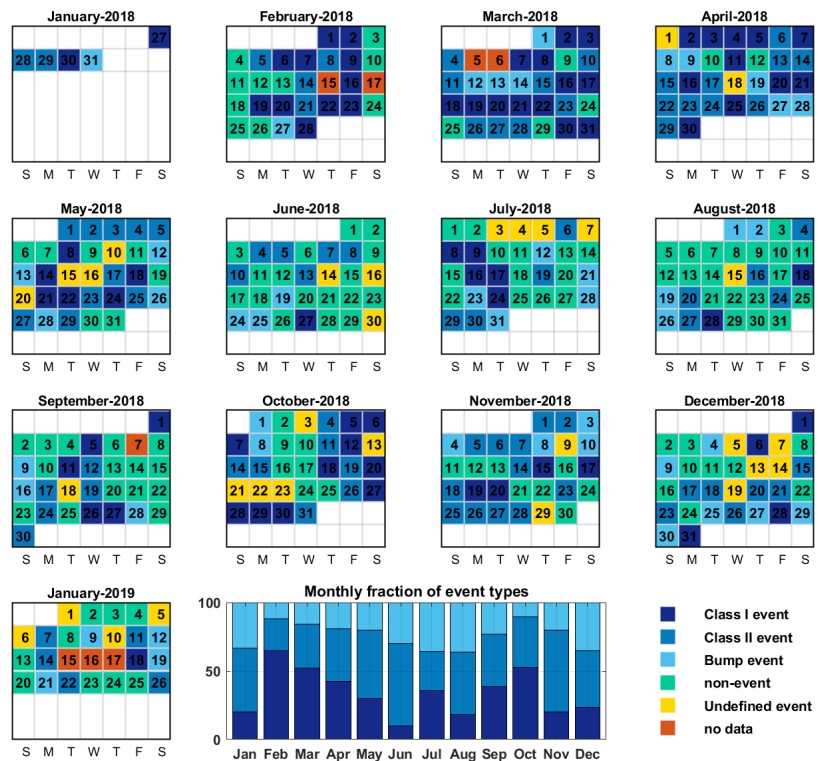

Figure 4. Calendar of daily event day classification between January 27, 2018 –January 26, 2019 and the monthly fraction of NPF event classes.





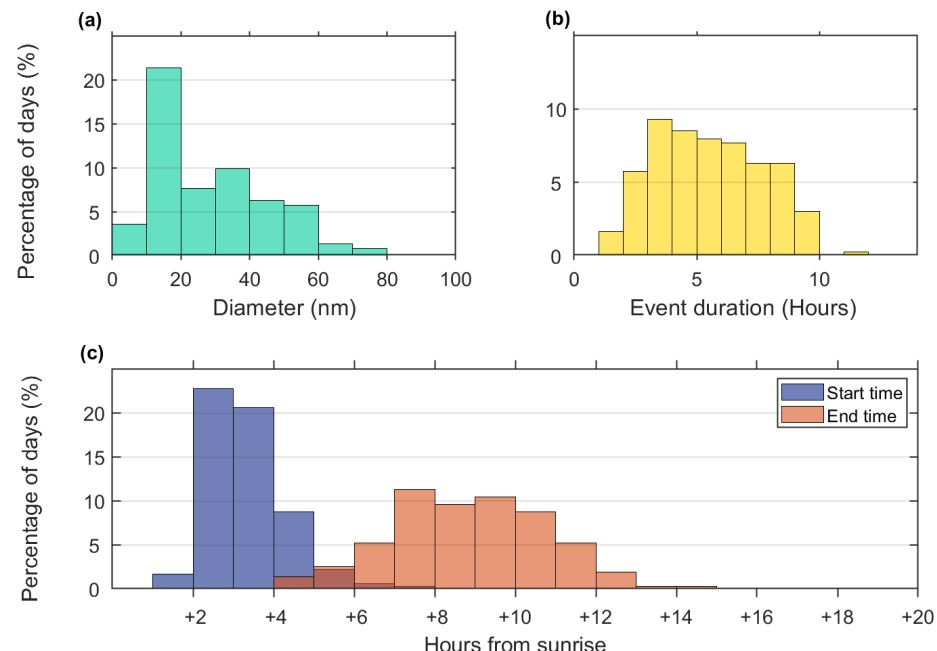

Figure 5. Percentage histograms showing the frequency distribution of (a) NPF events growing to a certain diameter, (b) NPF event duration, and (c) the event start and end times from sunrise.

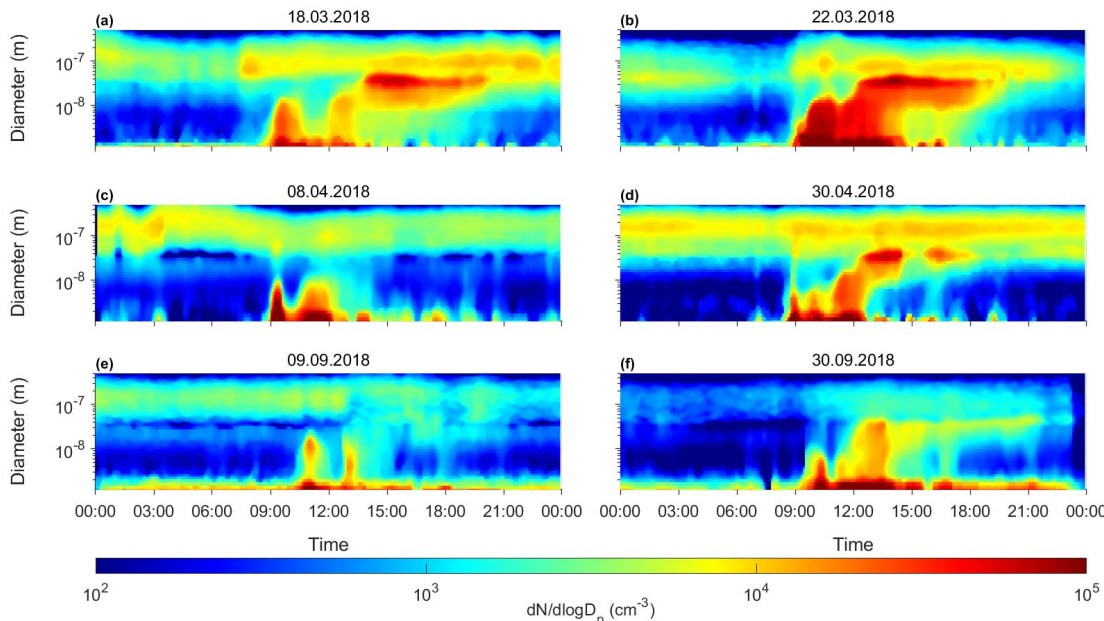

Figure 6. Examples of days with multiple nucleation events: (a) March 18, 2018 (b) March 22, 2018 (c) April 8, 2019, (d) April 30, 2018, (e) September 9, 2018 and (f) September 30, 2018.



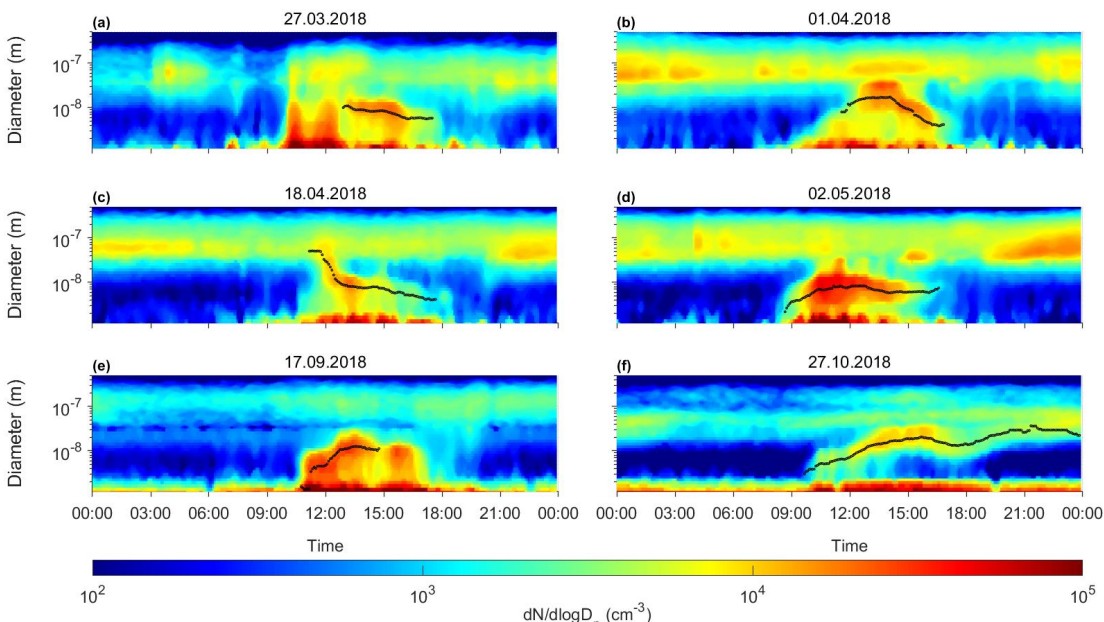

Figure 7. Example of days showing a decreasing mode diameter: (a) March 27, 2018, (b) April 1, 2018, (c) April 18, 2018, (d) May 2, 2018, (e) September 17, 2018 and (f) October 27, 2018 in local time. The mode diameter is plotted as black circular markers.

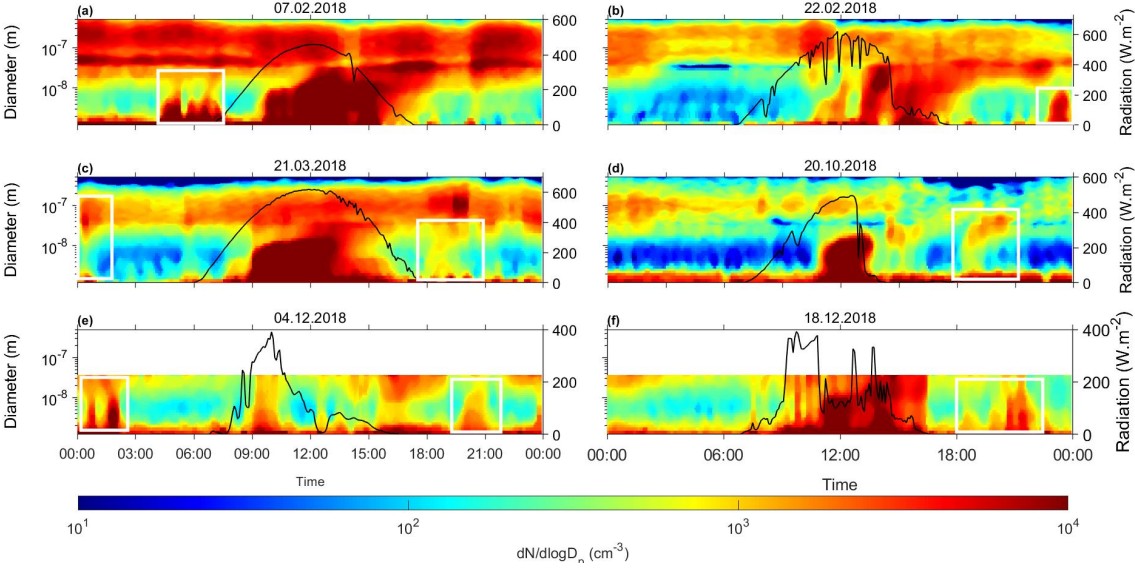

Figure 8. Examples of days with nighttime clustering and growth marked with white rectangles: (a) February 7, 2018, (b) February 22, 2018, (c) March 21, 2018, (d) October 20, 2018, (e) December 04, 2018 and (f) December 18, 2018. The black line is the solar radiation (W.m$^{-2}$) which can be read from the right axis. Note the difference in the color scale used in this figure in comparison to figures 6 and 7.





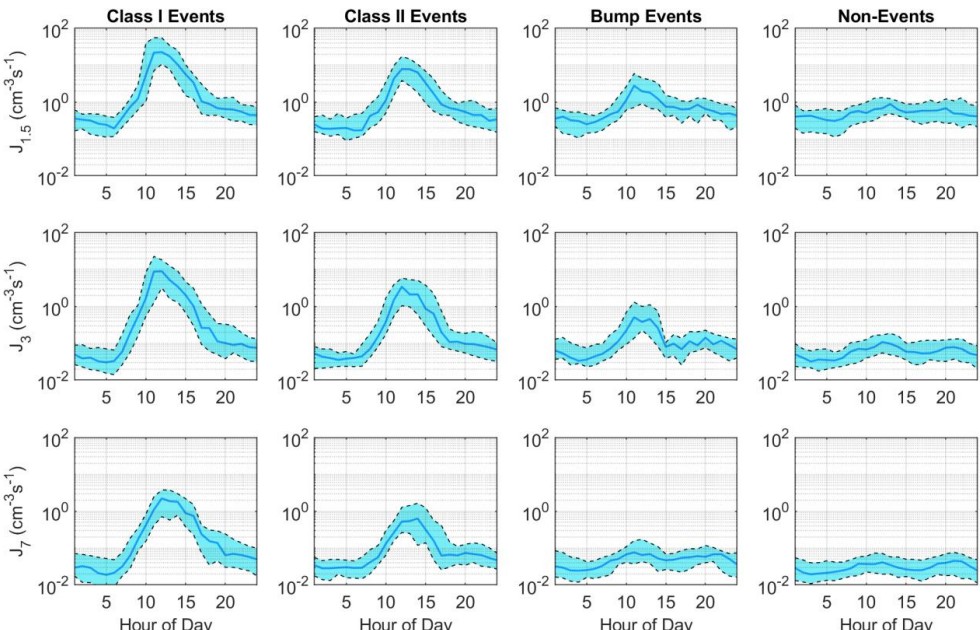

Figure 9. Diurnal variation of J1.5 (top), J3 (middle) and J7 (bottom) during class I, class II, bump events and non-events. Shaded areas represent the 25th and 75th percentile bounds while the solid line represents the median.





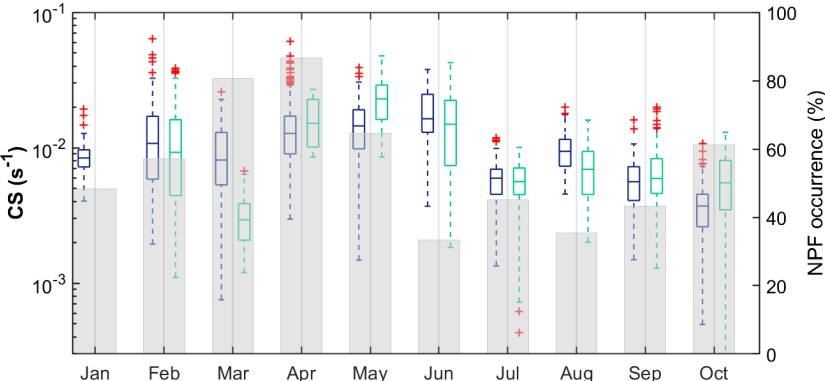

Figure 10. Monthly variation of condensation sink (s⁻¹) during event (blue) and non-event (green) days using data corresponding to global radiation is greater than 50 W.m⁻². The bottom and top edges of the box plots indicate the 25th and 75th percentiles, respectively. The central mark indicates the median. The whiskers extend to the most extreme data points not considered outliers, and the outliers are plotted individually using the '+' symbol. Data presented have hourly time resolution. The shaded grey bars represent the monthly NPF percent occurrence

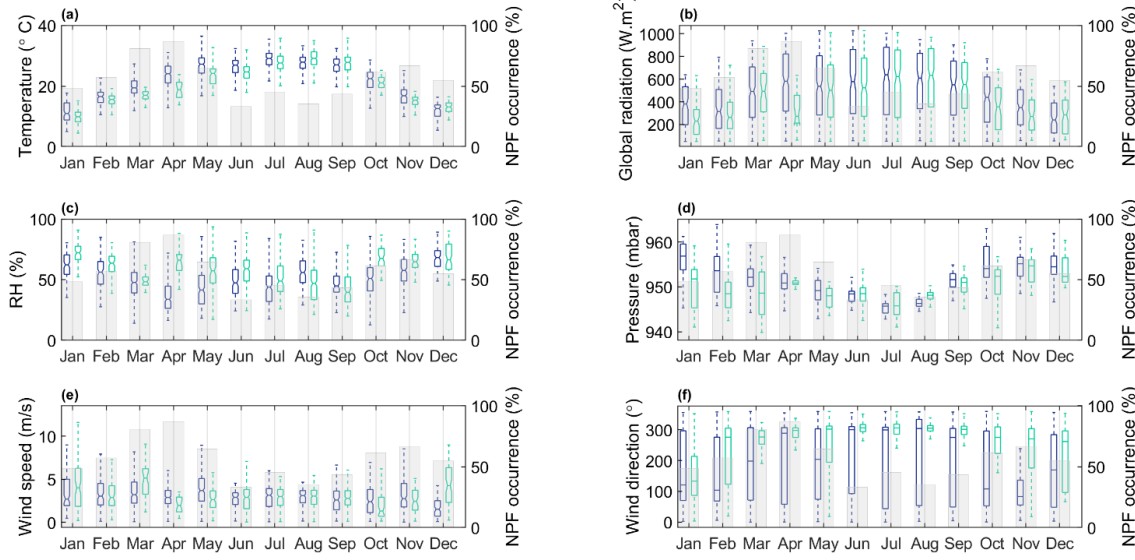

Figure 11. Monthly variation of metrological parameters during event (blue) and non-event (green) days: (a) temperature, (b) global radiation, (c) relative humidity, (d) pressure, (e) wind speed, and (f) wind direction using data corresponding to global radiation is greater than 50 W.m⁻². The bottom and top edges of the box plots indicate the 25th and 75th percentiles, respectively. The central mark indicates the median. The whiskers extend to the extreme data points not considered outliers. Data presented have hourly time resolution. The shaded grey bars represent the monthly NPF percent occurrence





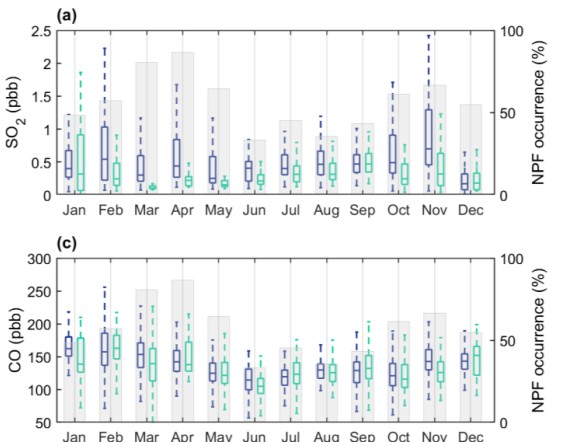

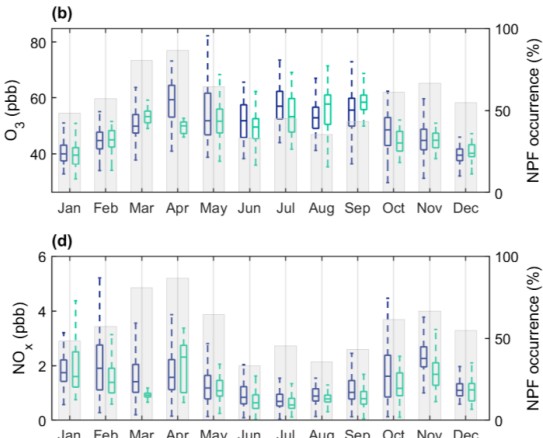

Figure 12. Monthly variation of trace gases during event (blue) and non-event (green) days: (a) sulfur dioxide, (b) ozone, (c) carbon monoxide, and (d) nitrogen oxide using data corresponding to global radiation is greater than 50 W.m⁻². The shaded grey bars represent the monthly NPF percent occurrence. The bottom and top edges of the box plots indicate the 25th and 75th percentiles, respectively. The central mark indicates the median. The whiskers extend to the most extreme data points not considered outliers. Data presented have hourly time resolution.



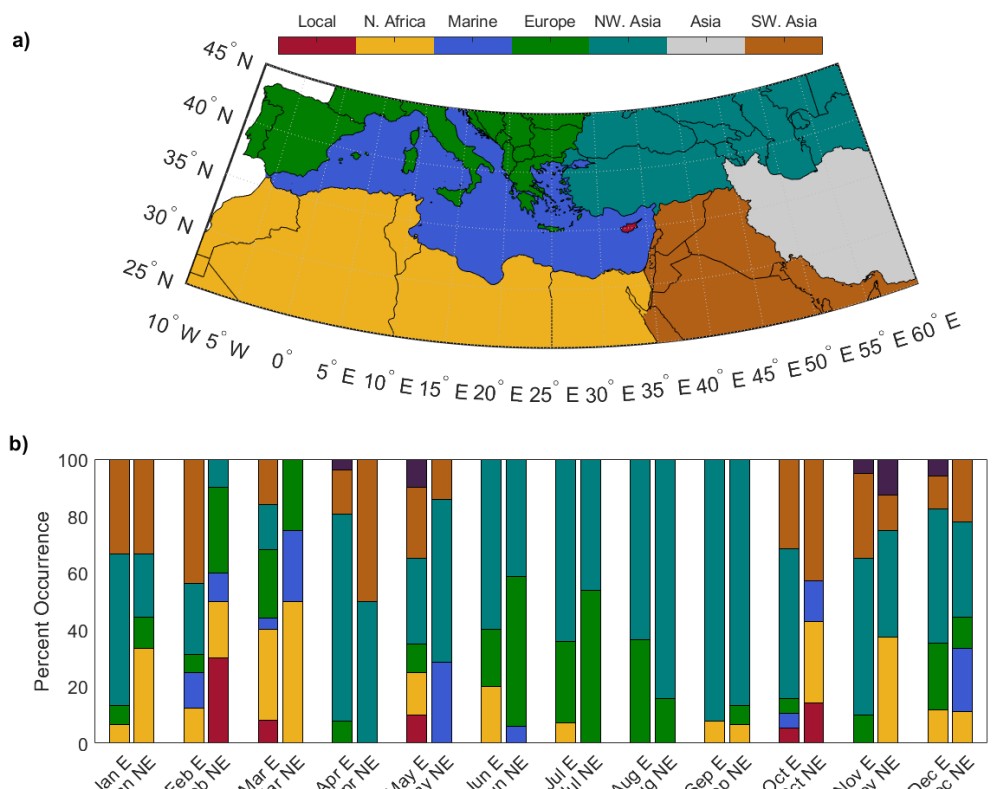

Figure 13. a) The source regions of air masses reaching CAO used for the air mass sector analysis. The map was plotted using The Climate Data Toolbox for MATLAB (Greene et al., 2019). The IHO World Sea Areas v3 were used to retrieve the boundaries of the Mediterranean Sea (Flanders Marine Institute, 2019). Note that marine areas other than the Mediterranean Sea were considered part of the continental sectors and that the NW. Asia and SW. Asia sectors are with respect to Cyprus location. b) Monthly variation of air mass origin arriving at CAO at 8:00 a.m. during event (E) and non-event days (NE). There are no air masses originating from Asia sector because those are obscured by terrain height.





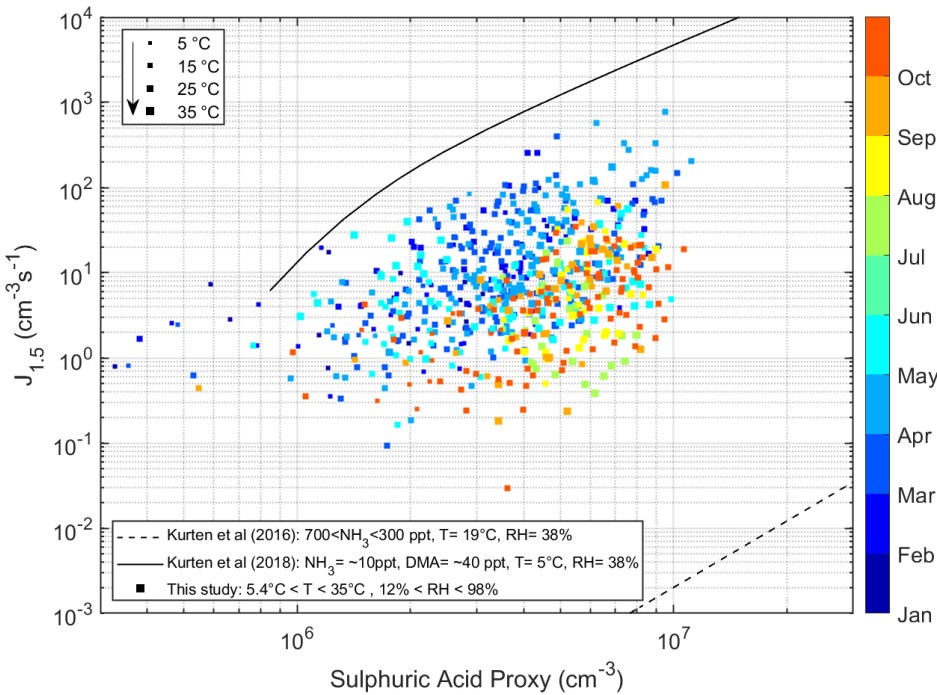

Figure 14. $J_{1.5}$ versus sulfuric acid proxy concentrations color-coded by the month of the year. Data presented have hourly time resolution.



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
