# Peer review of "Towards understanding the mechanisms of new particle formation"

_Atmospheric Chemistry and Physics, 2020_

## Referee Comment (RC1) · Anonymous Referee #2 · 5 Jan 2021

The study by Baalbaki and co-workers presents one year of measurements performed at a rural background station in Cyprus, with the aim of documenting the occurrence and characteristics of new particle formation (NPF) at this site located in the Eastern Mediterranean and Middle East (EMME) region and influenced by air masses of various origins. The paper is well written and presents new and interesting observations of the process in a poorly documented area. I therefore recommend the publication of this study in ACP, but I suggest to consider the comments listed below prior to publication. These comments mainly concern the main text, but, when necessary, the abstract and conclusion should be modified accordingly.

Comment 1: P1, L19: "*located at the crossroads of three continents*": In addition, I would mention explicitly (but very briefly!) the diversity of the air masses sampled at this site (marine vs. continental, desert vs. anthropized vs. forest areas) that results from this particular location, because I think it is an important specificity of the site that should be underlined already in the abstract.

Comment 2: P3, L97: "*which is the key gas phase precursor for cluster formation*": I would suggest modulating this assertion (and also that on P14, L535), since there are certain environments where this species does not, a priori, play a determining or limiting role in the early NPF process, and this seems to be the case in particular at CAO (P15, L580: "*sulfuric acid was not the limiting factor for NPF occurrence*")!

Comment 3: P4, L113-114: "*it is distant from any major pollution sources*": I would specify here "local sources", since it is indicated a few lines later (L119) that the site is frequently under the influence of air masses which transport pollutants from Turkey and Europe.

Comment 4: P4-5, L141-142: Were the authors able to evaluate the impact of this setup modification on the continuity/comparability of the measurements? Also, in addition to the effect on the CPC activation efficiency that is discussed here, RH of the sample flow also certainly affects the detection efficiency of the PSM itself (as a result of water-DEG interaction, Kangasluoma et al, 2013).

Comment 5: P5, L161, nCNC data handling: I would suggest specifying the number of size bins used in the main text. Also, I wonder why such a high number of classes is needed for this study, and, more importantly, I wonder about the relevance of such fine size classes, which width is of the same order or less than the uncertainty on particle sizing related to the particle chemical composition and/or charge (Kangasluoma et al., 2014; 2016). Wouldn't a single class, which could be used both in the calculation of $J_{1.5}$ and for cluster concentration analysis (e.g. 1.5-2.4 nm), be sufficient and less uncertain?

Comment 6: P5, L180, full PSD:

- Unless I am mistaken, the final temporal resolution of the composite PSD is not indicated, could the authors please specify it? It is indicated in Section 2.3 of the Supplement "*Here, we mainly use 1 hour resolution data for the presented analysis*": if this resolution is indeed used, isn't it too coarse to capture the rapid changes in cluster/particle concentration during NPF events? If the constraint comes from the PSM data, wouldn't the use of a single bin make it possible to increase this temporal resolution with more confidence?
- It is not clear to me at what stage the correction for particle hygroscopicity was applied: to the composite PSD after correcting NAIS concentrations? In this case, it would mean that the NAIS correction was made by comparing ambient PSD from NAIS and dry PSD for SMPS?
- Based on the values reported in the Supplement, it seems that seasonal means of kappa have been used to evaluate the particle HGF. I imagine that beyond the seasonal variability, there may be a variability of kappa related to the origin of the air masses; in the study of Holmgren et al. (2014), it was for instance shown that such variability was not obvious, but has it been investigated in the case of CAO?

Comment 7: P6, L206-207: "*spectrums of total particles (both neutral and charged) are usually less ambiguous to classify than charged particle spectra (ion mode of NAIS)*": could the authors explain why the total particles spectra less ambiguous to classify than that of ions? Is it related to the strength of the events? If so, this makes me wonder why the GR calculation is based on measurements of charged particles (P7, L224); also, why the negative ion mode data were specifically chosen for this calculation (instead of positive mode data, or both)?

Comment 8: P8, L255-256: "*but also includes the formation pathway via stabilized Criegee Intermediates*": I do not think it is the case for the proxy developed for rural sites and recalled in Eq. 6.

Comment 9: P8, L257: Air mass origin analysis: Although it has been described in previous studies, I would suggest to say a few words on the method used for the air mass classification, and especially on the definition of the geographical sectors, which are otherwise numerous!

Comment 10: P9, Fig. 2: Although the variability on the measurements is shown in Fig. 2, it is not described, and seasonal variations appear to be discussed primarily on the basis of monthly medians. I think it would be interesting to complete/modulate the conclusions in the light of this variability, which "unusual" behaviour in some months is also worth discussing (e.g. variability of the cluster concentration in February). I would also suggest adding a grid to make the figure easier to read. These recommendations also apply to Fig. 10, 11 and 12 and their analysis.

Comment 11: P9, L291-292: "*cluster mode and nucleation mode particles had roughly a similar pattern, with the highest concentrations during the spring followed by the autumn and a clear drop during the summer*": as indicated, there are "*rough*" similarities in the seasonal variations of the cluster/particle concentration in these two size classes. However, I do not think that the proposed brief description recalled above is appropriate since it does not seem to me to correspond to either situation. For example, fall/winter levels are similar or even higher than spring levels in the case of cluster concentration, while they are closer to summer levels in the case of the nucleation mode concentration. In fact, I think it would be particularly interesting to discuss these differences since it is indicated right after (L292-293) that concentrations in both size ranges "*can be directly linked to NPF activity*".

Comment 12: P9, L296: "*higher emission during spring*": does the authors refer to the emission of particles or gaseous precursors?

Comment 13: P9, L304: "*with the highest values recorded between 9:00 and 15:00 am and the maximum at 11:00*": The maximum concentrations in the two size classes appear at the same time while we would expect a chronology if they are related to consecutive NPF stages: is this only related to the hourly resolution of the data shown in the figure?

Comment 14: P11, L361-362: multiple events: how are these events taken into account in the statistics shown in Fig. 5? Only the first one observed, or the most intense one of the series, each one individually, or all the events are considered as a whole?

Comment 15: P11, L392: "*high dust loading (translated to a high condensation sink)*": did the authors estimate the increase in CS caused by the presence of dust compared to a dust-free day? These "big" particles have a definite impact on the PM but I would be curious to know their real impact on the CS!

Comment 16: P11, L401: particle formation rates:

- I think a figure (same type as Fig. 2, 10, 11 and 12) would make it easier to visualize the variations of the particle formation rates (Table 1 could be kept in the Supplement). To limit the number of figures in the paper, Fig. 6, 7, and 8 could possibly be transferred to the Supplement;
- Also, I do not understand how the values presented in Tables 1 and S4 were calculated. What does "daily" data mean, since it seems that in each case the reported statistics were calculated within the event duration?

- Finally, could the authors comment on the different seasonal patterns which are observed for the particle formation rates at different sizes (maybe in connection with the differences observed between Fig. 2.a and b, see Comment 11)?

Comment 17: P12, L441-442: "*The seasonal variation of the CS followed the seasonal pattern of the accumulation mode particles*": I pretty much agree with that, and I think therefore that in their present form, the descriptions of Fig. 2.d and 10 sound a bit contradictory:

- P9, L296-297: "*The accumulation mode had its maximum during the summer, except during July*"
- P12, L442-443: "*with highest values calculated for winter and spring, and lowest for summer and autumn*"

More generally, I suggest moving the CS discussion to the next section. This analysis currently sits astride sections 3.3 and 3.4, and I think that CS is more a "*driving atmospheric parameter*" of NPF than a "*specific parameter*" of the process.

Comment 18: P13, L450: I would suggest to explicitly refer to particle formation rates (also on P14, L534).

Comment 19: P13, L457-460: A possible explanation for the fact that the CS is not systematically lower on NPF event days could be that, as observed at some mountain sites (e.g. Sellegri et al. 2019), the sources and sinks (i.e., CS) of NPF precursors share the same origin, and in this case the CS is not necessarily a limiting factor.

Comment 20: P13, L486-487: "*NPF occurred largely at lower wind speeds and local north easterly winds, which is the direction where the main agglomerations and livestock farming lands are situated*":

- The use of box plot does not seem to me perfectly adapted to the analysis of wind direction, especially since the values close to the extremes 0 and 360° correspond to situations that are in fact similar. I think that the use of wind roses such as those shown in Fig. S11 is much more appropriate;
- Considering the variability of wind speed, it does not seem obvious to me from Fig. 11.e that weak winds are particularly favorable to NPF;
- Regarding wind direction, I would suggest to slightly rephrase this sentence. I might be wrong but it means for me that the majority of the events take place in north easterly wind, whereas there are months, and especially the months with the highest frequencies (March-May), when both sectors (i.e. north east and north west) seem to be almost equally represented on event days. Based on my understanding, I would say that NPF occurs in both north easterly and westerly winds but with a probability of occurrence which seems to be definitely higher in north easterly winds;
- What do the authors imply by indicating the presence of anthropogenic activities in the northeast sector? That there could be a local influence of these activities on the occurrence of NPF? Did the authors analyse whether there was a significant difference in the type of event (i.e. bump vs. class I or II) in the northeast and northwest sectors which could support this hypothesis?

Comment 21: P13, L494-495: In light of the reported observations, I would slightly balance this statement and rather say that $SO_2$ / $H_2SO_4$ "*cannot explain alone the seasonal pattern of NPF*".

Comment 22: P14, effect of air mass origin:

- L511: shouldn't it be *7* source regions?
- L513-514: shouldn't it be *months*?
- Although the marine sector does not appear to be one of the most frequent source regions in Fig. 13.b, I would assume that due to the insular nature of the station, "marine conditions" are

part of the history of a certain number of air masses sampled at CAO (as also suggested on P4, L121), and could therefore often influence the occurrence of NPF at this site. I think that a couple of sentences on this subject could enrich the discussion.

Comment 23: P14, L525-527: *"The increase in PM levels during these months could be a limiting factor for NPF. Indeed, accumulation mode particles and thus CS were the highest during the summer (except for July)"*. Based on Fig. 10, it is not obvious to me that the CS is higher in summer, and this is not what is indicated on P12, L441-443 (e.g. August vs February-May, also see Comment 17). I also wonder about the sequence of these two sentences ("*Indeed*") since no clear relationship between the CS (based on sub-700 nm particles) and the occurrence of NPF could be evidenced at CAO.

I would therefore suggest to remove the second sentence and simply mention that the actual CS could be higher in summer during episodes of high of PM levels, possibly contributing to lower NPF frequencies and less frequent particle growth during this time of the year.

Comment 24: P15, L555: I fully agree with this hypothesis, which explains why the individual analysis of the different atmospheric variables does not make it possible to highlight a preponderant role of one or the other of these variables. Without necessarily considering the use of "heavy" statistical approaches, have the authors tried to study some combinations of these variables, which behaviour could be more contrasted between event and non-event days and give clues about their combined role in the occurrence of NPF?

Comment 25: P27-28, Figs. 10-12: I would suggest to move the gray bars in the background to increase the clarity of the figures.

References

Holmgren, H., Sellegri, K., Hervo, M., Rose, C., Freney, E., Villani, P., and Laj, P.: Hygroscopic properties and mixing state of aerosol measured at the high-altitude site Puy de Dôme (1465 m a.s.l.), France, Atmos. Chem. Phys., 14, 9537–9554, https://doi.org/10.5194/acp-14-9537-2014, 2014.

J. Kangasluoma, H. Junninen, K. Lehtipalo, J. Mikkilä, J. Vanhanen, M. Attoui, M. Sipilä, D. Worsnop, M. Kulmala and T. Petäjä: Remarks on Ion Generation for CPC Detection Efficiency Studies in Sub-3-nm Size Range, Aerosol Science and Technology, 556-563, DOI: 10.1080/02786826.2013.773393, 2013.

Kangasluoma, J., Kuang, C., Wimmer, D., Rissanen, M., Lehtipalo, K., Ehn, M., Worsnop, D., Wang, J., Kulmala, M. and Petäjä, T.: Sub-3 nm particle size and composition dependent response of a nano-CPC battery, Atmospheric 765 Measurement Techniques, 7(3), 689-700, doi:10.5194/amt-7-689-2014, 2014.

Kangasluoma, J., Samodurov, A., Attoui, M., Franchin, A., Junninen, H., Korhonen, F., Kurtén, T., Vehkamäki, H., Sipilä, M., Lehtipalo, K., Worsnop, D., Petäjä, T. and Kulmala, M.: Heterogeneous Nucleation onto Ions and Neutralized Ions: Insights into Sign-Preference, The Journal of Physical Chemistry C, 120(13), 7444-7450, 770 doi:10.1021/acs.jpcc.6b01779, 2016.

Sellegri, K., Rose, C., Marinoni, A., Lupi, A., Wiedensohler, A., Andrade, M., Bonasoni, P. and Laj, P.: New Particle Formation: A Review of Ground-Based Observations at Mountain Research Stations, Atmosphere, 10(9), 493, doi:10.3390/atmos10090493, 2019.

---

## Referee Comment (RC2) · Anonymous Referee #1 · 13 Jan 2021

The manuscript titled "Towards understanding the mechanisms of new particle formation in the Eastern Mediterranean" presents yearlong observations of NPF events at a rural background location in Cyprus. Observations are based on various instrumentation providing information about NPF events since the early cluster sizes. These are very important observations in the poorly presented in the literature region of East Mediterranean and Middle East and it is worth being published after some minor revisions. I think however that the title is rather misleading since the manuscript is focused on the description of NPF events in Cyprus and their general characteristics and it does not contribute to actually understanding the underlying processes governing the formation of atmospheric particles and therefore I recommend a more modest title.

General comments

[Figure]

The authors are only briefly describing observations of NPF events during periods that desert dust was present in the atmosphere. Although it has been pointed out that mixed conditions of dust and pollution may result to the formation of new particles even under conditions with high preexisting aerosol loadings, the observations reported in the literature are scarce and only in few locations around the world. During the study period, 37 out of the 50 dust days were categorized as NPF days. This is an extraordinary figure and these events should have been prioritized in their analysis, given that under dust conditions it is more possible to have an NPF event (74%) compared to the average situation (57%). On the contrary, the authors choose not to present a single event. Even if it is chosen to present these events in a separate research article, the intention of the present work to introduce the scientific community to a novel location under the EMME atmospheric conditions which are greatly affected by the presence of desert dust makes the presentation of such NPF events in more detail necessary.

Another general comment has to do with the presentation of the driving parameters of NPF in the atmosphere of EMME. The authors have available a great set of complementary measurements to examine which atmospheric conditions favor or suppress NPF. The authors choose to present annual variability of each parameter rather than utilizing simple statistical tests to explore possible correlations. Visual inspection of event vs non-event conditions is not enough to contribute to the understanding the mechanisms of NPF and I would like to see some more in depth analysis such as PMF, PCA or simply regression analysis, for instance of cluster mode number concentration vs the various atmospheric components.

Specific comments.

L. 101: The most populated island in the Mediterranean is Sicily, Cyprus is the third most populous.

L. 103: Also Isreal to the southeast.

L. 139: How were the data prior to June 2018 treated with regard to activation efficien-

cies distortion?

L. 208: A reference is needed here to support this statement.

L. 231: The start and end time are not fully described here, more details should be given.

L. 350: The calendar does not contribute to the discussion of the results, it rather occupies a great extent of the given page. I would prefer to move the diurnal patterns from Supplementary material next to annual variations and remove the calendar.

L. 387: Since there are only few references of dust relevant NPF event in the literature, these 37 events should be described in more detail and compared to dust free days. At least an example of such possible events should be given.

L. 405: How do you support your hypothesis? This is highly speculative.

L.407: I would like to see all these information about Js in a Figure like 2 or 9.

L. 408: How have the J values reported in Table1 been calculated, ie from average daily J values, maximum daily values, average values during event duration or something else?

L. 433: Once again a figure for GR would be nice here.

L. 493: However, during the same period, SO2 concentrations are much higher during events than during non events, it seems that the SO2 abundance does make a difference.

L. 515: What compounds could that be? Such an assumption may be investigated looking for instance at SO2 charts for the region.

Table 1: Remove the period punctuation mark from the units of J.

[Figure]

---

## Author Comment (AC1) · 19 Apr 2021

**Review of "Towards understanding the mechanisms of new particle formation in the Eastern Mediterranean" by Anonymous Referee #1**

The manuscript titled "Towards understanding the mechanisms of new particle formation in the Eastern Mediterranean" presents yearlong observations of NPF events at a rural background location in Cyprus. Observations are based on various instrumentation providing information about NPF events since the early cluster sizes. These are very important observations in the poorly presented in the literature region of East Mediterranean and Middle East and it is worth being published after some minor revisions.

I think however that the title is rather misleading since the manuscript is focused on the description of NPF events in Cyprus and their general characteristics and it does not contribute to actually understanding the underlying processes governing the formation of atmospheric particles and therefore I recommend a more modest title.

We thank the reviewer for the constructive comments. We believe that this manuscript not only unveils the characteristics of NPF but also provides insight into the underlying processes governing the formation of atmospheric particles. We analyzed the seasonality of sulfuric acid, NPF sinks, and formation rates, which are all important factors for understanding the processes of NPF. We also show that the formation rates measured at this sit cannot be explained by ammonia-sulfuric acid nucleation alone. That said, we agree with the reviewer that a more modest title is more suitable. As such, we have changed the title of the manuscript to: "*Towards understanding the characteristics of new particle formation in the Eastern Mediterranean*".

We provide our point-to-point replies to the general and specific comments below. The reviewer comments are in black, our replies are in green, the text before adjustment is in orange, and the adjustments made to the manuscript are in blue. The corresponding changes are noted in the manuscript by track changes. The referred line and figure numbers in these replies denote the new ones in the revised manuscript and Supplementary Data. All references are provided at the end of the replies.

**General comments**

**Comment 1:** The authors are only briefly describing observations of NPF events during periods that desert dust was present in the atmosphere. Although it has been pointed out that mixed conditions of dust and pollution may result to the formation of new particles even under conditions with high preexisting aerosol loadings, the observations reported in the literature are scarce and only in few locations around the world. During the study period, 37 out of the 50 dust days were categorized as NPF days. This is an extraordinary figure and these events should have been prioritized in their analysis, given that under dust conditions it is more possible to have an NPF event (74%) compared to the average situation (57%). On the contrary, the authors choose not to present a single event. Even if it is chosen to present these events in a separate research article, the intention of the present work to introduce the scientific community to a novel location under the EMME atmospheric conditions which are greatly affected by the presence of desert dust makes the presentation of such NPF events in more detail necessary.

We agree with the reviewer on the importance of presenting dust events in more depth. As such, we have added the subsequent section to the manuscript.

**Addition to manuscript:**

Figure 10 shows the temporal variation of  $PM_{10\cdot2.5}$ ,  $PM_{2.5}$ , and particle number size distribution measured during three of the dust episodes with ±5 days window before and after the dust episode. NPF took place at high dust loadings, and there is no obvious threshold for the dust loading above which NPF does not occur. In addition, the formation rates (Figure S9) and growth rates (Figure S10) between NPF event days not affected by high dust loading, and NPF event days affected by high dust loadings seem to be comparable. J7 was slightly higher on days affected by high dust loading, but this could be related to the lower number of dust cases compared with the non-dust cases. High dust loadings can affect NPF in opposing ways. On the one hand, it can suppress photochemical processes by scavenging reactive gases and condensable vapors (De Reus et al., 2000;Ndour et al., 2009). On the other hand, it can provide particles that can act as a site for heterogeneous photochemistry promoting the formation of gaseous OH radicals, which initiate the conversion of SO2 to

 $H_2SO_4$  (Dupart et al., 2012; Nie et al., 2014). However, a clear association between high dust loading and NPF was not found from the data set presented here.

---

## Author Comment (AC2) · 19 Apr 2021

**Review of Baalbaki et al "Towards understanding the mechanisms of new particle formation in the Eastern Mediterranean" by Anonymous Referee #2**

The study by Baalbaki and co-workers presents one year of measurements performed at a rural background station in Cyprus, with the aim of documenting the occurrence and characteristics of new particle formation (NPF) at this site located in the Eastern Mediterranean and Middle East (EMME) region and influenced by air masses of various origins. The paper is well written and presents new and interesting observations of the process in a poorly documented area. I therefore recommend the publication of this study in ACP, but I suggest to consider the comments listed below prior to publication. These comments mainly concern the main text, but, when necessary, the abstract and conclusion should be modified accordingly.

We thank the reviewer for the constructive comments. We provide our point-to-point replies to the general and specific comments below. The reviewer comments are in black, our replies are in green, the text before adjustment is in orange, and the adjustments made to the manuscript are in blue. The referred line and figure numbers in these replies denote the new ones in the revised manuscript and Supplementary Data. All references are provided at the end of the replies.
We have additionally changed the title of the manuscript based on the recommendation of reviewer 1. The title now reads: "*Towards understanding the characteristics of new particle formation in the Eastern Mediterranean*".

**Comment 1**: P1, L19: "*located at the crossroads of three continents*": In addition, I would mention explicitly (but very briefly!) the diversity of the air masses sampled at this site (marine vs. continental, desert vs. anthropized vs. forest areas) that results from this particular location, because I think it is an important specificity of the site that should be underlined already in the abstract.

Modified as suggested.

Before correction:
Here, we study NPF in Cyprus, an Eastern Mediterranean country located at the crossroads of three continents.

After correction:
Here, we study NPF in Cyprus, an Eastern Mediterranean country located at the crossroads of three continents and affected by diverse air masses originating from continental, maritime and desert-dust source areas.

**Comment 2**: P3, L97: "*which is the key gas phase precursor for cluster formation*": I would suggest modulating this assertion (and also that on P14, L535), since there are certain environments where this species does not, a priori, play a determining or limiting role in the early NPF process, and this seems to be the case in particular at CAO (P15, L580: "*sulfuric acid was not the limiting factor for NPF occurrence*")!

Both sentences were rephrased as per below.

Before correction:
- We further explore the role of sulfuric acid, which is the key gas phase precursor for cluster formation, and other atmospheric variables in initiating NPF at this site.
- While sulfuric acid (H2SO4) is the main nucleating species in the atmosphere

After correction:
- We further explore the role of sulfuric acid, which is one of the key gas phase precursors for cluster formation, and other atmospheric variables that can potentially be associated with NPF at this site.
- While sulfuric acid ($H_2SO_4$) is considered as one of the main nucleating species in the atmosphere

**Comment 3:** P4, L113-114: "*it is distant from any major pollution sources*": I would specify here "local sources", since it is indicated a few lines later (L119) that the site is frequently under the influence of air masses which transport pollutants from Turkey and Europe.

The sentence now reads as follows: it is not directly affected by any major local pollution source.

**Comment 4:** P4-5, L141-142: Were the authors able to evaluate the impact of this setup modification on the continuity/comparability of the measurements? Also, in addition to the effect on the CPC activation efficiency that is discussed here, RH of the sample flow also certainly affects the detection efficiency of the PSM itself (as a result of water-DEG interaction, (Kangasluoma et al., 2013)).

We noticed that the original text did not reflect properly the actual problem faced with the CPC. The problem that arose during the summer was that high water vapor content in the sampled air was mixing with the butanol of the CPC resulting in sub-saturated conditions. Thus, the CPC was not able to activate particles at all and was measuring zero. This problem did not occur before the summer. We updated the text accordingly

Before correction:

From June 2018 onwards, the PSM was additionally equipped with a diluter to reduce the humidity of the sampled air. This procedure was necessary because the water content of the air at the measurement site was too high, and it affected the activation efficiency inside the CPC and therefore distorted the size distribution measurements for the smallest sizes. Further information about the diluter design and operation can be found in the supplementary information (SI) Sect. 2.2.

After correction:

From June 2018 onwards, the nCNC was additionally equipped with a diluter to reduce the humidity of the sampled air. This procedure was necessary because the water content of the air at the measurement site was too high. The water present in the sample air was mixed with butanol inside the CPC of the nCNC and rendered it measuring zeros. Further information about the diluter design, its operation and effect on the data can be found in the supplementary information (SI) Sect. 2.2.

That said, we discuss below the effects of the addition of the diluter on the PSM measurements. This discussion was added to the supplementary material.

The addition of the diluter has three different effects on the PSM measurements. The first effect is related to the penetration efficiency or line losses inside the diluter piece. The diluter's penetration efficiency was characterized in the laboratory and was found to be similar to that of the core-sampling inlet, which was used earlier in the study, so this effect is negligible. The second effect is related to possibly decreasing the water content in the sampled aerosols and thus making them smaller. However, we cannot correct for this effect because the hygroscopicity/dehydration of sub-3nm particles is not known. The third effect is related to the activation efficiency of particles at lower sample RH. Increased water content of the sample enhances both the DEG-water activation and DEG-aerosol-water activation. Since background zero measurements for the PSM were performed three times a day with filtered sample air, the effect of adding the diluter on the DEG-water interaction was indirectly monitored. The DEG-water counts were reduced after the addition of the diluter as can be seen from Figure S2 but they were mostly in the range of 0 to 10, which is within normal operation conditions of the PSM. Concerning the DEG-aerosol-water activation, the uncertainty due to changing RH ranges between 0-0.3 nm on the PSM cutoff, and is smaller than the uncertainty due to change in particle composition (0-1 nm) (Kangasluoma et al., 2013), which also cannot be controlled.

[Figure]

Figure S2. PSM counts at maximum saturator flow during zero measurements with filtered sampled air.

**Comment 5:** P5, L161, nCNC data handling: I would suggest specifying the number of size bins used in the main text. Also, I wonder why such a high number of classes is needed for this study, and, more importantly, I wonder about the relevance of such fine size classes, which width is of the same order or less than the uncertainty on particle sizing related to the particle chemical composition and/or charge (Kangasluoma et al., 2014;Kangasluoma et al., 2016). Wouldn't a single class, which could be used both in the calculation of $J_{1.5}$ and for cluster concentration analysis (e.g. 1.5-2.4 nm), be sufficient and less uncertain?

The number of size bins was specified in the main text as suggested by the reviewer.
The following diameters were used in the inversion: 1.1 nm, 1.3nm, 1.5 nm, and 2.4 nm .

The reasoning behind the number of the size bins used is as follows:
For the kernel inversion method, only four diameters between 1.1 and 2.4 nm were used in the inversion, which is the usual number of size bins used in this least square inversion method. The number of bins is not considered high as the PSM actually scans ~120 different sizes while varying the saturator flow rate between 0.1 and 1.3 lpm. One size class could have been used in the analysis as it has less uncertainty as the reviewer suggests but the finer size bins give indication about the evolution of the size distribution in sub-3nm sizes and provide overlapping measurements with the lower bins of the NAIS. For the EM inversion method, 50 kernels representing 50 detection efficiency curves were used. Unlike the kernel method, the EM method does not have limitations of the number of size bins used ($d_j$) because it is a statistical method that repeats an expectation step and a maximization step until convergence. On the contrary, a larger number of size bins (J) can reduce the errors in approximating the integral using the discrete sum in the following equations of the expectation and minimization steps, respectively:

$$R_{i,j} = \frac{n_j \times \eta(s_i, d_j) \times \Delta d_j}{\sum_{j=1}^{j} n_j \times \eta(s_i, d_j) \times \Delta d_j}$$

$$n_j = \frac{\sum_{i=1}^{I} R_{i,j}}{\sum_{l=1}^{I} \eta(s_i, d_j) \times \Delta d_j}$$

Where $R_{i,j}$ is the detected total particle concentration at the $i^{th}$ saturator flow rate and $j^{th}$ size bin, n is the particle size distribution function, $\eta$ is the is the overall detection efficiency determined by s, the saturator flow rate, and d the particle diameter, and $\Delta d$ is the length of each size bin.

**Comment 6:** P5, L180, full PSD:
- Unless I am mistaken, the final temporal resolution of the composite PSD is not indicated, could the authors please specify it? It is indicated in Section 2.3 of the Supplement "*Here, we mainly use 1 hour resolution data for the presented analysis*": if this resolution is indeed used, isn't it too coarse to capture the rapid changes in cluster/particle concentration during NPF events? If the constraint comes from the PSM data, wouldn't the use of a single bin make it possible to increase this temporal resolution with more confidence?

The time resolution is now added. The temporal resolution of the PSD was five minutes. All the surface plots presented in the manuscript have five minutes time resolution and the growth rate calculations were done using the five-minute data. A one-hour time resolution was chosen for the remainder of the analysis for three main reasons. The first reason is related to the type of analysis presented in this manuscript. We are interested in NPF events, which by definition are events longer than one hour in time. In our analysis we present the diurnal variation but the main focus is on the seasonality of NPF which requires averaging out the fluctuations in high resolution data. The second reason is that the gas data ($O_3$, CO, $NO_x$ and $SO_2$) and PM data were only available in hourly resolution. The third reason is that hourly PSM data have less uncertainty. The comparison between the EM and the kernel inversion method showed that these two methods have less variation among each other when hourly medians are compared. Thus, the uncertainty of the reported concentrations and thus J-values are lower in the hourly data. The PSM data had originally had 5-minute resolution after the inversion. Indeed, as the reviewer suggests, using a single size bin from the PSM would increase the temporal resolution of PSM data but only if the PSM was operated in fixed mode. In our case the PSM was operated in scanning mode, thus the temporal resolution would not increase even if we use only one size bin for inverting the data.

After correction:
Complementary meteorological data (temperature, relative humidity, solar radiation, rainfall, pressure, wind speed and wind direction) were measured with a time resolution of five minutes at an elevation of 10 m from the ground in the nearby village of Xyliatos (35.0140917 N, 33.0492028 E), 2.85 km from the measurement site. Air pollutants (ozone, carbon monoxide, nitrogen oxides, sulfur oxide, PM10 and PM2.5) were measured at the collocated EMEP station ~20 m from the main measurement container, and these data had a time resolution of one hour.

Full Particle Size Distribution (PSD): The data from the three particle sizing instruments were used to reconstruct the full particle size distribution with a temporal resolution of 5 minutes between 1.1 and 736 nm

- It is not clear to me at what stage the correction for particle hygroscopicity was applied: to the composite PSD after correcting NAIS concentrations? In this case, it would mean that the NAIS correction was made by comparing ambient PSD from NAIS and dry PSD for SMPS?

The hygroscopicity correction was applied to the SMPS data before making the composite PSD. This has been now been clarified by rearranging the text as per below.

Before correction:
*Full Particle Size Distribution (PSD):* The data from the three particle sizing instruments were used to reconstruct the full particle size distribution between 1.1 and 736 nm (nCNC: 1.1 to 2.4 nm; NAIS particle 181 mode: 2.4 to 30 nm; SMPS: 30 to 736 nm). However, since the NAIS is known to overestimate concentrations in particle mode, the overlapping measurement range with the SMPS was used to further correct the NAIS data assuming that the NAIS overestimate concentrations uniformly over the whole measurement range, which is a reasonable assumption for old NAIS models based on calibration results (Gagné et al., 2011;Kangasluoma et al., 2020). Additionally, the SMPS measured dry aerosol particle number distributions, which can differ considerably from the ambient aerosol particle number size distribution. Thus, we back-calculated the distribution at ambient conditions from the dry distribution using the hygroscopicity model of Petters and 188 Kreidenweis (2007) and mean kappa values. Additional information about these calculations and its effect on sink calculations are presented in Sect. 4 of the SI material. The full PSD using the distribution at ambient conditions was reconstructed up to 1500 nm. This does not imply that the measurement range was extended to 1500 nm, but rather that now we account for particles that were originally of sizes up to 1500 nm but were dried to sizes below 740 nm in the SMPS sampling line.

After correction:
*Full Particle Size Distribution (PSD):* The data from the three particle sizing instruments were used to reconstruct the full particle size distribution with a temporal resolution of 5 minutes between 1.1 and 736 nm (nCNC: 1.1 to 2.4 nm; NAIS particle mode: 2.4 to 30 nm; SMPS: 30 to 736 nm). However, the SMPS measured dry aerosol particle number distributions, which can differ from the ambient aerosol particle number size distribution. Thus, we back-calculated the distribution of the SMPS at ambient conditions from the dry distribution using the

hygroscopicity model of Petters and Kreidenweis (2007) and mean kappa values. Additional information about these calculations and its effect on sink calculations are presented in Sect. four of the SI material. The SMPS distribution at ambient conditions was reconstructed up to 1500 nm. This does not imply that the measurement range was extended to 1500 nm, but rather that now we account for particles that were originally of sizes up to 1500 nm but were dried to sizes below 740 nm in the SMPS sampling line. Additionally, since the NAIS is known to overestimate concentrations in particle mode, the overlapping measurement range with the SMPS was used to further correct the NAIS data assuming that the NAIS overestimate concentrations uniformly over the whole measurement range, which is a reasonable assumption for old NAIS models based on calibration results (Gagné et al., 2011;Kangasluoma et al., 2020).

- Based on the values reported in the Supplement, it seems that seasonal means of kappa have been used to evaluate the particle HGF. I imagine that beyond the seasonal variability, there may be a variability of kappa related to the origin of the air masses; in the study of Holmgren et al. (2014), it was for instance shown that such variability was not obvious, but has it been investigated in the case of CAO?

We acknowledge that understanding how the origin of air masses affect particle composition, and thus hygroscopicity expressed by kappa, is an important subject. However, this analysis is beyond the scope of our study, which is focused on NPF monthly and seasonal variability. Here we used the hygroscopicity data to reduce the uncertainties related to the different pretreatment of the aerosols prior to sampling by the different instruments. The variability in kappa throughout the whole HTDMA measurement period was less than 0.1 as expressed by the standard deviation. Thus, the use of seasonal kappa values was justified. That said, a study focused on the particle hygroscopicity observed at CAO will follow in future work.

**Comment 7:** P6, L206-207: "*spectrums of total particles (both neutral and charged) are usually less ambiguous to classify than charged particle spectra (ion mode of NAIS)*": could the authors explain why the total particles spectra less ambiguous to classify than that of ions? Is it related to the strength of the events? If so, this makes me wonder why the GR calculation is based on measurements of charged particles (P7, L224); also, why the negative ion mode data were specifically chosen for this calculation (instead of positive mode data, or both)?

Even though ion-induced and ion-mediated processes could play an important role in nucleation mainly in the early formation steps (Kirkby et al., 2016;He et al., 2021;Jokinen et al., 2018), neutral nucleation processes continue to dominate atmospheric observations (Kulmala et al., 2013;Kontkanen et al., 2013;Wagner et al., 2017). In our observations, the spectra of total particles are less ambiguous to classify, not only because events are stronger in the neutral channel but also because the charged spectra usually show higher concentrations at bigger particle sizes than lower sizes which makes it hard to verify that nucleation started from sub 3nm sizes. An example of this typical behavior is shown in figure R1 for two different dates using the NAIS ion and particle data.

The main text was adjusted accordingly.

Before correction:
In addition, spectrums of total particles (both neutral and charged) are usually less ambiguous to classify than charged particle spectra (ion mode of NAIS), and the classification of event days may be different if one only looks at these charged spectrums.

After correction:
In addition, spectra of total particles (both neutral and charged) are usually easier to visually classify than those corresponding to charged particles (measured by the ion mode of NAIS) because atmospheric nucleation is dominated by neutral processes (Kontkanen et al., 2013;Kulmala et al., 2013;Wagner et al., 2017). In addition, the concentration of the growing mode in the charged spectra is lower for the smaller particles, and increases with diameter as the probability of cluster ions attaching to the growing neutral particles increases (Gonser et al., 2014). Thus, it could be visually difficult to determine if particle nucleation starts from the smallest sizes when

looking at charged spectra only. In contrast, one should not neglect looking at charged spectra because those might show sign preference or ion induced nucleation events (Rose et al., 2018).

[Figure]

Figure R1. An example of two dates (right and left) showing contrasting spectra from the NAIS negative ion mode (top), positive ion mode (middle) and total particles mode (bottom)

Concerning the growth rate calculations, the reviewer's concern about choosing only negative ion data for the calculations is justified as earlier observations show that total particles exhibit enhanced growth rates at diameters below 15 nm (Gonser et al., 2014). Thus, we now present the growth rate analysis for negative ions, positive ions and total particles as shown in Figure 13. We also modified the analysis according to the new results.

[Figure]

Figure 13. Monthly variation of growth rates during NPF events in three size ranges: (a) <3nm (b) 3-7nm and (c) 7-20nm. The central marks indicate the median, the bottom and top edges of the big box indicate the 25th and 75th percentiles, respectively. The whiskers extend' to the most extreme data points not considered outliers, and the outliers are plotted individually using the '+' symbol .The numbers above the box plot represent the number of data points within each boxplot. Black boxes represent the total particles (neutral+charges), blue boxes represent negative ions and red boxes represent positive ions.

Before correction:

We report size-segregated growth rates between 1.5 and 3 nm (GR 1.5–3), between 3 and 7 nm 418 (GR 3–7), and between 7 and 20 nm (GR 7–20) as recommended by Kulmala et al. (2012) (Table 3). The median growth rates in these size ranges were 2.0, 4.8, and 7.4 nm hr-1, respectively. These GRs are higher than those reported for a rural boreal environment (1.9, 3.8 and 4.3 nm hr-1, respectively) (Yli-Juuti et al., 2011), but within the range of GRs reported for 12 European sites (Manninen et al., 2010 cf. Figure S8). The increase in the growth rate with an increasing particle size is a typical feature in the sub-20 nm size range because condensational growth is more favorable as the particle size increases and the Kelvin effect decreases (Manninen et al., 2010). Additionally, we calculated the growth rates between 3 and 25 nm (GR 3-25) for the purpose of making comparison with additional studies. Similar to J3, the median GR 3-25 calculated here (6.3 nm hr-1) was comparable to the range reported for urban or rural sites with urban influence (4.7 – 7.7 nm hr-1), whereas rural sites usually have a GR 3-25 below 4 nm hr-1. In regard to the seasonal variability, we did not find a clear pattern in GR 7-20 or GR 3-25. In contrast, GR 1.5–3 and GR 3–7 were generally higher during the summer months, which could be associated to the higher fraction of bump events with respect to other event types. As discussed in Sect. 3.3, these bump events are characterized by a burst of particles within a short period of time, which would translate to higher growth rates.

After correction:

We report size-segregated growth rates between 1.5 and 3 nm (GR $_{1.5-3}$), between 3 and 7 nm (GR $_{3-7}$), and between 7 and 20 nm (GR $_{7-20}$) as recommended by Kulmala et al. (2012) for negatively charged ions, positively charged ions and total particles (charged + neutral) (Figure 13). The growth rates of total particles were higher than that of the charged fraction, which is in agreement with earlier studies showing enhanced growth rates in the neutral channel at diameters below 15 nm (Gonser et al., 2014;Manninen et al., 2009;Rohan Jayaratne et al., 2016). This behavior has been explained by Gonser et al. (2014) whom provided a conceptual model of the influence of cluster ion recombination and attachment at different stages of particle nucleation and growth. The seasonal behavior of the growth rates was also distinct. In the sub 3 nm range, the negative ions growth rates had similar median values across the year except during July, which had higher growth rates whereas the positive polarity had notable increase in the growth rates in the summer month. The difference in the growth rates at these cluster sizes suggests that the ion induced NPF processes are more important in the positive channel. In the 3-7nm size range, there was no clear seasonal pattern except that June had the highest growth rates in the negative and positive mode while the month exhibiting the highest growth rates in the total particle mode were February and June. In the 7-20 nm size range, the growth rates exhibited a clear seasonality in all channels with a peak in February and another broad peak during the summer month. The GR increased with an increasing particle size, which is a typical feature in the sub-20 nm size range because condensational growth is more favorable as the particle size increases and the Kelvin effect decreases (Manninen et al., 2010). The median growth rates in the three size ranges (calculated from the daily means of the three channels) were 3.7, 9.2, and 11.7 nm hr$^{-1}$, respectively. These GRs are higher than those reported for a rural boreal environment (1.9, 3.8 and 4.3 nm hr$^{-1}$, respectively) (Yli-Juuti et al., 2011). In comparison to other studies, the ion mode GR reported here is on the higher range of GRs measured at 12 European sites (Manninen et al., 2010 cf. Figure S8).. The high growth rates reported here could be associated to the high fraction of bump events. As discussed in Sect. 3.3, these events are characterized by a burst of particles within a short period of time, which would translate to higher growth rates.

**Comment 8:** P8, L255-256: "*but also includes the formation pathway via stabilized Criegee Intermediates*": I do not think it is the case for the proxy developed for rural sites and recalled in Eq. 6.

Indeed, this is not the case for the proxy developed for rural sites. The text was describing the general advances in the new proxy presented by (Dada et al., 2020). We modified the text to clarify this difference.

Before correction:
This proxy does not only consider the formation of H2SO4 from SO2 via OH oxidation and the loss towards pre-existing particles (condensation sink), but also includes the formation pathway via stabilized Criegee Intermediates and loss towards atmospheric clustering starting from H2SO4 dimer formation

After correction:
This proxy does not only consider the formation of $H_2SO_4$ from $SO_2$ via OH oxidation and loss of $H_2SO_4$ onto pre-existing particles (condensation sink), but it also includes loss of $H_2SO_4$ via atmospheric clustering starting from $H_2SO_4$ dimer formation.

**Comment 9:** P8, L257: Air mass origin analysis: Although it has been described in previous studies, I would suggest to say a few words on the method used for the air mass classification, and especially on the definition of the geographical sectors, which are otherwise numerous!

A detailed explanation about the air mass classification and the definition of the geographical sector was added to the main text as shown below:

Before correction:
Seven source regions were identified similar to the ones presented by Pikridas et al. (2018) except that in our analysis, the West Turkey sector was merged to the NW Asia sector.

After correction:
In general, a retroplume was attributed to a region in the case that this had a PES value above 0.9 ns kg$^{-1}$. The classification scheme of the source regions took into consideration dominant air mass paths shown by Pikridas et al. (2018) and the different sources of PM with characteristic chemical fingerprint. As a result, the predominant northerly air masses were categorized into Europe and NW Asia (namely Turkey), assuming different emissions

related to SO$_2$. N Africa and SW Asia are both source areas of dust particles but with distinct emission levels, with the former being associated with more elevated concentrations. The Asia sector was distinguished to point out that air masses from this specific source region scarcely reach the receptor site, while the source region Local refers to stagnant conditions. Last, the Marine sector is associated with the lowest levels of ambient PM. In total, seven source regions were identified similar to the ones presented by Pikridas et al. (2018), except that in our analysis the West Turkey sector was merged into the NW Asia sector.

**Comment 10:** P9, Fig. 2: Although the variability on the measurements is shown in Fig. 2, it is not described, and seasonal variations appear to be discussed primarily on the basis of monthly medians. I think it would be interesting to complete/modulate the conclusions in the light of this variability, which "unusual" behaviour in some months is also worth discussing (e.g. variability of the cluster concentration in February). I would also suggest adding a grid to make the figure easier to read. These recommendations also apply to Fig. 10, 11 and 12 and their analysis.

We agree with the reviewer that it is important to discuss the variability of the particle number concentration beyond the monthly medians. We have adjusted the text as per below taking into account comment 11.

We have also added the grid to the figures as suggested.

Before correction:

A clear seasonal pattern is depicted which is distinct across the different particle modes. The cluster mode and nucleation mode particles had roughly a similar pattern, with the highest concentrations during the spring followed by the autumn and a clear drop during the summer. The cluster and nucleation mode concentrations can be directly linked to the NPF activity, especially in sites where direct emissions of particles having these size ranges are minimal, which is the case for our site. The Aitken mode exhibited higher concentrations during the spring months followed by a decreasing pattern, which could either suggest more growth from NPF to Aitken sizes or higher emission during spring. The accumulation mode had its maximum during the summer, except during July which did not follow the pattern of other months. Previous long-term measurements of PM$_{2.5}$ at this site have a similar pattern with higher concentrations during the warm period of the year and minimum during winter (Pikridas et al., 2018). This maximum during the summer is mainly explained by the enhanced transport of polluted air masses from the north sector, combined with the lack of precipitation and overall dry conditions during Eastern Mediterranean summer (Pikridas et al., 2018).

After correction:

A clear seasonal pattern is depicted which is distinct across the different particle modes. The cluster mode particles had two peaks, one in spring and another in autumn with a clear drop during the summer. The monthly box plots also show high variability in daily concentrations throughout most months, except August and September. This variability seems highest during February. The nucleation mode particles were also highest during the spring and lowest during the summer. The autumn concentrations did not exhibit another peak but were rather similar to the summer concentrations (in terms of median values) but they exhibited higher variability. The cluster and nucleation mode concentrations can be directly linked to the NPF activity, especially in sites where direct emissions of particles having these size ranges are minimal, which is the case for our site. While the high concentrations of cluster mode particles during spring was associated with high concentrations of nucleation mode particles, this did not hold for autumn, which might indicate that condensable vapors were not as available to grow the particles to nucleation size. The Aitken mode exhibited higher concentrations during the spring months followed by a decreasing pattern, which could either suggest more growth from NPF to Aitken sizes or higher emission/transportation of primary particles during spring. The accumulation mode had its maximum during the warm months, except during July which did not follow the pattern of other months. Previous long-term measurements of PM$_{2.5}$ at this site have a similar pattern with higher concentrations during the warm period of the year and minimum during winter (Pikridas et al., 2018). This maximum during the summer is mainly explained by the enhanced transport of polluted air masses from the north sector, combined with the lack of precipitation and overall dry conditions during Eastern Mediterranean summer (Pikridas et al., 2018). Last, it is worth mentioning that during February, the concentrations of particles in all modes did not follow the overall trend. It exhibited lower concentrations of cluster, nucleation, and Aitken mode particles and higher concentration of accumulation mode particles than the nearby month.

**Comment 11:** P9, L291-292: "*cluster mode and nucleation mode particles had roughly a similar pattern, with the highest concentrations during the spring followed by the autumn and a clear drop during the summer*": as indicated, there are "*rough*" similarities in the seasonal variations of the cluster/particle concentration in these two size classes. However, I do not think that the proposed brief description recalled above is appropriate since it does not seem to me to correspond to either situation. For example, fall/winter levels are similar or even higher than spring levels in the case of cluster concentration, while they are closer to summer levels in the case of the nucleation mode concentration. In fact, I think it would be particularly interesting to discuss these differences since it is indicated right after (L292-293) that concentrations in both size ranges "*can be directly linked to NPF activity*".

We agree with the reviewer's concern. We have adjusted the text as shown in the replies to comment 10.

**Comment 12:** P9, L296: "*higher emission during spring*": does the authors refer to the emission of particles or gaseous precursors?

Here we refer to higher emission of particles. The text was adjusted to clarify this point.

Before correction:
The Aitken mode exhibited higher concentrations during the spring months followed by a decreasing pattern, which could either suggest more growth from NPF to Aitken sizes or higher emission during spring.

After correction:
The Aitken mode exhibited higher concentrations during the spring months followed by a decreasing pattern that could either suggest more growth from NPF to Aitken sizes or higher emission/transportation of particles during spring.

**Comment 13:** P9, L304: "*with the highest values recorded between 9:00 and 15:00 am and the maximum at 11:00*": The maximum concentrations in the two size classes appear at the same time while we would expect a chronology if they are related to consecutive NPF stages: is this only related to the hourly resolution of the data shown in the figure?

Indeed, the two size classes seem to appear at the same time because of the hourly time resolution which hides the small time delay in the onset between the two sizes (Figure R4). The following sentence was added to the main text:
There was a slight time difference between the appearance of the cluster mode particles that of the nucleation mode particles which can only be seen in the 5-min data

[Figure]

Figure R2. The median diurnal cycle of cluster mode and nucleation mode presented with 5 min time resolution.

**Comment 14:** P11, L361-362: multiple events: how are these events taken into account in the statistics shown in Fig. 5? Only the first one observed, or the most intense one of the series, each one individually, or all the events are considered as a whole?

The multiple events were considered as one whole while computing the statistics shown in Fig. 5. This is now clarified in the text.

Addition to text:
In case of multiple events within a one-day window, the event start and end times were taken from the start of the first event until the end of the last event, respectively.

**Comment 15:** P11, L392: "*high dust loading (translated to a high condensation sink)*": did the authors estimate the increase in CS caused by the presence of dust compared to a dust-free day? These "big" particles have a definite impact on the PM but I would be curious to know their real impact on the CS!

It is indeed of great interest to know how much the high dust loading affects the CS, especially in this area of the world. Unfortunately, concurrent particle number concentration measurements above ~700 nm were not available to compute this effect. Thus, we were not able to rule out the probability that high PM loading in the summer is the cause of the low NPF occurrence.

**Comment 16:** P11, L401: particle formation rates:
- I think a figure (same type as Fig. 2, 10, 11 and 12) would make it easier to visualize the variations of the particle formation rates (Table 1 could be kept in the Supplement). To limit the number of figures in the paper, Fig. 6, 7, and 8 could possibly be transferred to the Supplement;

Formation rates are now presented as box plots. Instead of tables, we will alternatively place the data on Zenodo.

[Figure]

Figure 11. Monthly variation of particle formation rates during NPF events: (a) $J_{1.5}$, (b) $J_3$ and (c) $J_7$. The central marks indicate the median, the blue small boxes indicate the mean, the bottom and top edges of the big box indicate the 25th and 75th percentiles, respectively. The whiskers extend to the most extreme data points not considered outliers, and the outliers are plotted individually using the '+' symbol .The numbers above the box plot represent the number of data points within each boxplot.

- Also, I do not understand how the values presented in Tables 1 and S4 were calculated. What does "daily" data mean, since it seems that in each case the reported statistics were calculated within the event duration?

The difference between the data reported in Table 1 and that in Table S4 is in the data used to compute the monthly statistics. In case of Table 1, the daily averages were first calculated within the event duration, and then those were used to get the monthly statistics. Whereas in case of Table S4, the hourly data were directly used to calculate the monthly statistics. Since there is no consensus in the literature on how to compute the monthly statistics, we chose to present both tables to make future comparison to other studies possible. However, after replacing the tables with figures based on the requests of both reviewers, we will alternatively place the data on Zenodo.

**Comment 17:** P12, L441-442: "*The seasonal variation of the CS followed the seasonal pattern of the accumulation mode particles*": I pretty much agree with that, and I think therefore that in their present form, the descriptions of Fig. 2.d and 10 sound a bit contradictory:
- P9, L296-297: "*The accumulation mode had its maximum during the summer, except during July*"
- P12, L442-443: "*with highest values calculated for winter and spring, and lowest for summer and autumn*"

We thank the reviewer for spotting this discrepancy. The description on Fig. 2 was made by describing the month-to-month trend of the accumulation-mode particles while the description of the CS "*with highest values*

*calculated for winter and spring, and lowest for summer and autumn*" was made by comparing the seasonal CS values presented in table S5. We now realize that this table is biased because CS is not available for Nov and Dec. Thus, we deleted the table and the contradicting description it provided.

More generally, I suggest moving the CS discussion to the next section. This analysis currently sits astride sections 3.3 and 3.4, and I think that CS is more a "*driving atmospheric parameter*" of NPF than a "*specific parameter*" of the process.

CS section moved as suggested.

**Comment 18:** P13, L450: I would suggest to explicitly refer to particle formation rates (also on P14, L534).

Adjusted.

**Comment 19:** P13, L457-460: A possible explanation for the fact that the CS is not systematically lower on NPF event days could be that, as observed at some mountain sites (e.g. Sellegri et al. (2019), the sources and sinks (i.e., CS) of NPF precursors share the same origin, and in this case the CS is not necessarily a limiting factor.

We thank the reviewer for this insightful remark. We have added it to the text.

**Comment 20:** P13, L486-487: "*NPF occurred largely at lower wind speeds and local north easterly winds, which is the direction where the main agglomerations and livestock farming lands are situated*":
   - The use of box plot does not seem to me perfectly adapted to the analysis of wind direction, especially since the values close to the extremes 0 and 360° correspond to situations that are in fact similar. I think that the use of wind roses such as those shown in Fig. S11 is much more appropriate;

We deleted the subplots of wind speed and wind directions as suggested.

- Considering the variability of wind speed, it does not seem obvious to me from Fig. 11.e that weak winds are particularly favorable to NPF;

We adjusted the sentence under question.

Before correction:

NPF occurred largely at lower wind speeds and local north-easterly winds, which is the direction where the main agglomerations and livestock farming lands are situated (Figure 11e, 11f & S11).

After correction:

High wind speeds did not seem to prevent NPF but event days occurred mostly under low wind speeds (Figure S11).

- Regarding wind direction, I would suggest to slightly rephrase this sentence. I might be wrong but it means for me that the majority of the events take place in north easterly wind, whereas there are months, and especially the months with the highest frequencies (March-May), when both sectors (i.e. north east and north west) seem to be almost equally represented on event days. Based on my understanding, I would say that NPF occurs in both north easterly and westerly winds but with a probability of occurrence which seems to be definitely higher in north easterly winds;

We rephrased the sentence in question as suggested and added an extra figure supporting the discussion to the supplementary information showing the frequency of wind direction data divided by event types.

Before correction:
NPF occurred largely at lower wind speeds and local north-easterly winds, which is the direction where the main agglomerations and livestock farming lands are situated (Figure 11e, 11f & S11).

After correction:
With respect to wind speed, high wind speeds did not seem to prevent NPF but event days occurred mostly under low wind speeds (Figure S14). In terms of wind direction, NPF occurred mainly in when the wind was blowing

from west to east sector but with a frequency of occurrence which is higher in north easterly winds (Figure S14, S15, & S16). The north-easterly direction is the direction where the main agglomerations and livestock farming lands are situated. These local sources could be enhancing the occurrence of NPF but no direct relation was found between the north-easterly wind direction and specific event types (Figure S16). However, NPF class I events seem to be restricted to the west to east-southeast sector.

[Figure]

Figure S15. The frequency (a) and normalized frequency (b) of hourly wind direction data divided to 16 sectors color coded by the event classification.

[Figure]

Figure S16. The frequency (a) and normalized frequency (b) of daily wind direction data divided to 16 sectors color coded by the event classification

- What do the authors imply by indicating the presence of anthropogenic activities in the northeast sector? That there could be a local influence of these activities on the occurrence of NPF? Did the authors analyse whether there was a significant difference in the type of event (i.e. bump vs. class I or II) in the northeast and northwest sectors which could support this hypothesis?

We analyzed the type of events with respect to wind direction in Figure S15 & S16 but did not find a clear relationship between the event type and the northeasterly wind. However, we found that class I events seem to be restricted to the west to east-southeast sector (based on daily wind direction data).Thus, the text was adjusted accordingly as shown in the responses to comment 20.

**Comment 21:** P13, L494-495: In light of the reported observations, I would slightly balance this statement and rather say that SO$_2$ / H$_2$SO$_4$ "*cannot explain alone the seasonal pattern of NPF*".

Adjustment made as suggested.

**Comment 22:** P14, effect of air mass origin:
- L511: shouldn't it be *7* source regions?

While Fig. 17a shows seven source regions, only six regions are presented in Fig. 17b and in the text because there are no air masses originating from Asia sector as those are obscured by terrain height. This difference was clarified in the legend of Fig.17 and now further clarified in the main text

Addition to text:
Air masses did not originate from the Asia sector because they were obscured by the terrain height

- L513-514: shouldn't it be *months*?

Adjusted.

- Although the marine sector does not appear to be one of the most frequent source regions in Fig. 13.b, I would assume that due to the insular nature of the station, "marine conditions" are part of the history of a certain number of air masses sampled at CAO (as also suggested on P4, L121), and could therefore often influence the occurrence of NPF at this site. I think that a couple of sentences on this subject could enrich the discussion.

Discussion added as suggested.

Addition to text:
Pure marine air masses were not as frequent as other air masses. However, owing to the location of Cyprus, air masses from other continental source origins are expected to be influenced by marine conditions as they travel to our measurement site. Thus, we cannot exclude the potential marine effect on the occurrence of NPF.

**Comment 23:** P14, L525-527: "*The increase in PM levels during these months could be a limiting factor for NPF. Indeed, accumulation mode particles and thus CS were the highest during the summer (except for July)*". Based on Fig. 10, it is not obvious to me that the CS is higher in summer, and this is not what is indicated on P12, L441-443 (e.g. August vs February-May, also see Comment 17). I also wonder about the sequence of these two sentences ("*Indeed*") since no clear relationship between the CS (based on sub-700 nm particles) and the occurrence of NPF could be evidenced at CAO.
I would therefore suggest to remove the second sentence and simply mention that the actual CS could be higher in summer during episodes of high of PM levels, possibly contributing to lower NPF frequencies and less frequent particle growth during this time of the year.

The sentence causing contradiction with other parts was deleted as suggested.

**Comment 24:** P15, L555: I fully agree with this hypothesis, which explains why the individual analysis of the different atmospheric variables does not make it possible to highlight a preponderant role of one or the other of these variables. Without necessarily considering the use of "heavy" statistical approaches, have the authors tried to study some combinations of these variables, which behavior could be more contrasted between event and non-event days and give clues about their combined role in the occurrence of NPF?

An in depth statistical analysis was made and presented in the responses to reviewer 1 comment 2.

**Comment 25:** P27-28, Figs. 10-12: I would suggest to move the gray bars in the background to increase the clarity of the figures.

The figures were adjusted and the box plots are now on top of the grey bars.  (figs 10-12 are now Figs 14-16)

[revised manuscript text omitted]